# Modality independent or modality specific? Common computations underlie confidence judgements in visual and auditory decisions

**Rebecca K. West**[1]*, **William J. Harrison**[1,2], **Natasha Matthews**[1], **Jason B. Mattingley**[1,2,3], **David K. Sewell**[1]

**1** School of Psychology, University of Queensland, Queensland, Australia, **2** Queensland Brain Institute, University of Queensland, Queensland, Australia, **3** Canadian Institute for Advanced Research, Toronto, Canada

* rebecca.west1@uq.net.au

**Data Availability Statement:** All data and code have been made publicly accessible at: https://

## Abstract

The mechanisms that enable humans to evaluate their confidence across a range of different decisions remain poorly understood. To bridge this gap in understanding, we used computational modelling to investigate the processes that underlie confidence judgements for perceptual decisions and the extent to which these computations are the same in the visual and auditory modalities. Participants completed two versions of a categorisation task with visual or auditory stimuli and made confidence judgements about their category decisions. In each modality, we varied both evidence strength, (i.e., the strength of the evidence for a particular category) and sensory uncertainty (i.e., the intensity of the sensory signal). We evaluated several classes of computational models which formalise the mapping of evidence strength and sensory uncertainty to confidence in different ways: 1) unscaled evidence strength models, 2) scaled evidence strength models, and 3) Bayesian models. Our model comparison results showed that across tasks and modalities, participants take evidence strength and sensory uncertainty into account in a way that is consistent with the scaled evidence strength class. Notably, the Bayesian class provided a relatively poor account of the data across modalities, particularly in the more complex categorisation task. Our findings suggest that a common process is used for evaluating confidence in perceptual decisions across domains, but that the parameter settings governing the process are tuned differently in each modality. Overall, our results highlight the impact of sensory uncertainty on confidence and the unity of metacognitive processing across sensory modalities.

## Author summary

In this study, we investigated the computational processes that describe how people derive a sense of confidence in their decisions. In particular, we used computational models to describe how decision confidence is generated from different stimulus features, specifically *evidence strength* and *sensory uncertainty*, and determined whether the same computations generalise to both visual and auditory decisions. We tested a range of different

github.com/rebeccakw/Visual_Auditory_
Confidence_Computations.

**Funding:** JBM was supported by a National Health and Medical Research Council (Australia) Investigator Grant (GNT2010141). WJH was supported by an Australian Research Council Discovery Early Career Researcher Award (DE190100136). The funders had no role in study design, data collection and analysis, decision to publish, or preparation of the manuscript.

**Competing interests:** The authors have declared that no competing interests exist.

computational models from three distinct theoretical classes, where each class of models instantiated different algorithmic hypotheses about the computations that are used to generate confidence. We found that a single class of models, in which confidence is derived from a subjective assessment of the strength of the evidence for a particular choice scaled by an estimate of sensory uncertainty, provided the best account of confidence for both visual and auditory decisions. Our findings suggest that the same type of algorithm is used for evaluating confidence across sensory modalities but that the '*settings*' (or *parameters*) of this process are fine-tuned within each modality.

## Introduction

Humans possess a remarkable ability to flexibly and reliably evaluate their uncertainty in a range of decisions. Whether reflecting on knowledge (e.g., How sure am I that the capital of Australia is Canberra?), perception (e.g., Is that a cyclist on the road ahead?), or value judgements (e.g., Do I want to buy the latest smart phone?), people have an awareness of the quality of their decisions, even in the absence of explicit feedback [1, 2]. This awareness arises from a self-monitoring reflective process, termed *metacognition*, which allows people to assess the quality of their internal cognitive operations [3].

Metacognition has been quantified experimentally using retrospective confidence judgements of performance. In a typical paradigm, an observer makes a task response, typically referred to as the *Type 1* or *first-order* decision, and then reports their level of confidence in that response, referred to as the *Type 2* or *second-order* decision [4]. Using confidence judgements, several studies have reported a high degree of association between observers' task accuracy and their confidence. These accurate metacognitive evaluations have been reported in a range of visual, auditory, tactile, memory, arithmetic, reading, spelling and general-knowledge tasks [5–16]. In many of these contexts, observers evaluate their uncertainty in tasks which contain just a single stream of information. In the natural environment, however, we typically evaluate information from several different sensory and cognitive domains at any one time. For example, when deciding whether to buy a particular pineapple at the grocery store, one might evaluate their confidence that the pineapple's colour, smell, and texture indicate that it is ripe.

Currently, little is known about the computations that support the diversity and flexibility with which people monitor their confidence. In particular, it is unclear whether the computations used to evaluate confidence are functionally and algorithmically related across different sensory and cognitive domains. To address this question, we compared metacognitive judgements across two sensory modalities, vision and audition. We chose to compare confidence across modalities as it allowed us to keep the decision context the same but vary the domain of information presentation. Thus, we could isolate any similarities (*modality-independent* processes) or differences (*modality-specific* processes) in the computations used to generate confidence.

### Modality-independent or modality-specific metacognition?

There are theoretical justifications for expecting metacognitive processes to be either modality-specific or modality-independent. Sensory information in different modalities has different features, and the different sensory systems respond selectively to those features. Modality-specific metacognitive processes can capitalise on this specialised encoding and maximise the accuracy of uncertainty monitoring within a modality. In addition, we do not rely on all of our

senses equally [17–20]. We may use optimal but computationally expensive processes for the senses we use most often, such as vision [21–23], and have more heuristic, efficient metacognitive processes for other modalities, for example audition.

Modality independent processes, on the other hand, may be equally advantageous. The sensory environment is inherently multi-modal in nature. In the real world we consistently filter out irrelevant information from certain modalities, make comparative judgements and integrate information across modalities. Transforming modality-specific sensory input into a modality-independent confidence variable would create a common currency for flexible comparison and integration of uncertainty across domains [24, 25].

To compare these competing accounts, we evaluated a number of existing models which formalise the computations involved in generating choice and confidence judgements and compared the ability of such models to account for human performance in visual and auditory categorisation tasks. We consider three broad classes of existing confidence models: 1) unscaled evidence strength models, 2) scaled evidence strength models, and 3) Bayesian models. We provide an overview of these confidence models below. To illustrate the similarities and differences between models, in the following sections we focus on their implementation in a two-alternative force choice task (2AFC) where an observer decides if a Gabor patch is rotated clockwise or counter-clockwise relative to horizontal (a category decision; see two example gratings in **Fig 1A**) and evaluates their confidence in that decision.

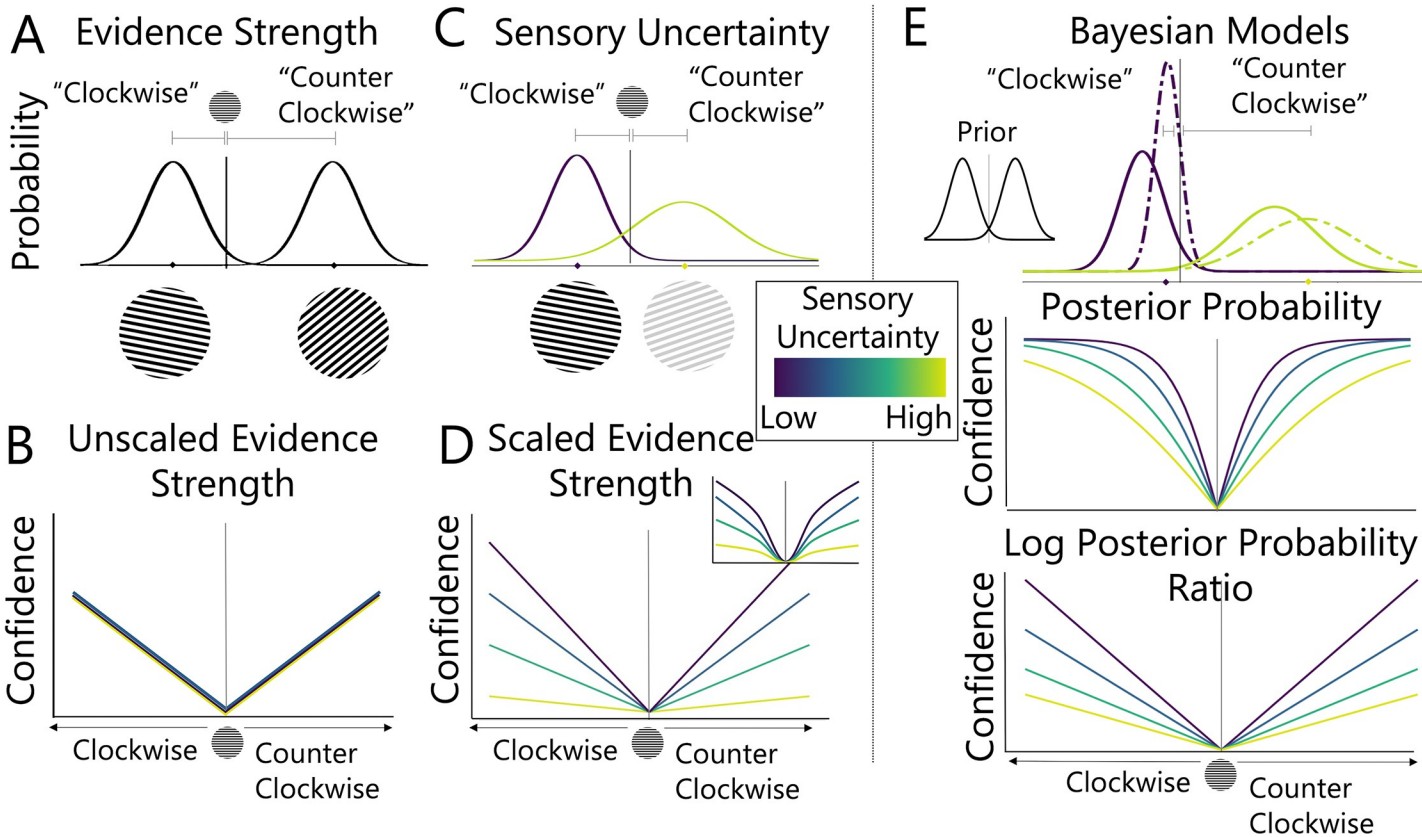

**Fig 1. Confidence Models Tested in the Current Study.** Illustration of (A) evidence strength and (C) sensory uncertainty concepts for confidence models. In this example, the probability distributions depict noisy representations of the orientation of a Gabor patch. Evidence strength varies with distance from the decision criterion and sensory uncertainty with the contrast of the Gabor. (B) Unscaled evidence strength models only factor in evidence strength. (D) Scaled evidence and (E) Bayesian models factor in both evidence strength and sensory uncertainty using different principles. Scaled evidence strength models assume a parametric relationship between confidence and uncertainty. Linear and quadratic (inset in D) examples are shown here. Bayesian models describe confidence in terms of the posterior probability or the log posterior probability ratio of competing outcomes.

As is common in the study of perception and cognition, the modelling approach we explored in this study is based on signal detection theory. We assume that the observer receives an incoming sensory signal, which is transformed into a *decision variable*. To make a binary decision, the observer compares the decision variable to a *decision criterion*. The confidence rating is then modelled as a secondary decision based on the same comparison process using additional *confidence criteria*. Different model classes make different theoretical assumptions about the computation of the decision variable.

## Evidence strength models

In the evidence strength models, the decision variable represents a direct estimate of the sensory signal, that is, a noisy representation of the stimulus feature of interest (see **Fig 1B**; where the stimulus feature of interest is the orientation of the Gabor patch). Evidence strength models posit that an observer's confidence depends on the strength of the evidence for a particular decision outcome, such that confidence is assumed to reflect the distance between the signal and the decision criterion. To illustrate this, see **Fig 1A** which shows two noisy measurements of the orientations of two Gabor patches with different distances from the category decision criterion. For numerical confidence ratings with *N* levels, *N*-1 criteria can be used to construct confidence bins that delineate the mapping between distance and confidence ratings. Evidence strength models can be further differentiated based on whether the positioning of confidence criteria takes *sensory uncertainty* into account (i.e., the amount of noise in the signal, which may vary, for example, with the contrast of the Gabor patch, see **Fig 1C**).

**Unscaled Evidence Strength.** According to unscaled evidence strength models, confidence reports depend only on the unsigned distance between the decision variable and the decision criterion [26–28]. In the case of the clockwise grating shown in **Fig 1C**, the evidence only weakly supports a clockwise decision. That is, the decision variable is close to the decision criterion, so a hypothetical observer may respond with low confidence. Unscaled evidence strength models do not independently quantify sensory uncertainty, such that a given distance will always be associated with the same level of confidence, regardless of other properties of the stimulus. These are perhaps the simplest models to reject because any empirical difference in observers' confidence judgments across levels of sensory uncertainty cannot be accounted for by the model.

**Scaled Evidence Strength.** According to scaled evidence strength models, confidence is assumed to be influenced not only by evidence strength but also by sensory uncertainty, the amount of noise in the signal. In scaled evidence strength models, an estimate of sensory uncertainty is used to scale evidence strength, for example, sensory uncertainty can be used to scale the confidence criteria used to delineate distance [29–32] or to normalise the distance itself [33, 34]. In effect this means that under conditions of high sensory uncertainty (light green line in **Fig 1C and 1D**), greater evidence strength is required to produce the same level of confidence as under conditions of low sensory uncertainty (purple line in **Fig 1C and 1D**). Several models fall into this class where different scaling rules can be applied. Adler and Ma (2018), for example, tested models where category and confidence response criteria were estimated as linear (depicted in main plot of **Fig 1D**) or quadratic (depicted in inset of **Fig 1D**) functions of sensory uncertainty. Here we consider category and confidence criteria that are linear and quadratic (and other) monotonic functions of sensory uncertainty, but we do not draw any theoretical distinctions between the different scaling rules.

## Bayesian models

Bayesian confidence models constitute a statistically optimal way of mapping both task and sensory information to confidence. In Bayesian models, observers combine knowledge about

the statistical structure of the task (the "prior") with the present sensory input (the "likeli-hood") to compute a posterior probability distribution over possible states of the stimulus [35– 41]. Bayesian models therefore differ from evidence strength models because the decision variable is a posterior probability distribution representing all possible states of the stimulus and their inferred probabilities. Confidence judgements are then derived from the probability metric. For the clockwise vs. counter clockwise decision illustrated in **Fig 1**, where $x$ is the measurement of the orientation of the Gabor patch, a Bayesian observer would report clock-wise whenever $p(clockwise|x)$ is above 0.5. The more extreme the posterior probability, the more confident the observer can be in their response. An alternative way of mapping posterior probability to confidence is to use the log posterior probability ratio, $\log\left(\frac{p(clockwise|x)}{p(counter\ clockwise|x)}\right)$, which is unbounded and allows for greater differences in confidence at different levels of sen-sory uncertainty (see bottom panel of **Fig 1E** [29, 33]).

## Model comparison

To date, formal comparisons of these different classes of confidence models have focused on the visual domain and have yielded conflicting results with some studies finding that visual confidence judgements are most consistent with Bayesian models [3, 15, 37, 42, 43] and others finding evidence for non-Bayesian models [3, 29, 33, 44, 45]. Thus, not only are there demon-strated inconsistencies in research findings within the visual domain, but it is also currently unclear how well the existing set of confidence models can account for confidence judgements across modalities. A formal investigation of the computations used to generate confidence across modalities and tasks is needed.

## The current study

The aim of the present study was to investigate the computations used to generate confidence judgements across sensory modalities by comparing the fits of different confidence models to data from different modalities and tasks (henceforth, all factorial combinations of modality and task are collectively referred to as *task-modality configurations*). We had participants undertake two versions of a categorisation task, one requiring decisions on visual stimuli and the other, decisions on auditory stimuli (see **Fig 2**). We equated task demands across modali-ties to allow us to distinguish differences in metacognitive processes from overall changes in task performance. We varied both evidence strength (i.e., the strength of the evidence for a particular category) and sensory uncertainty (i.e., the intensity of the sensory signal) in each modality. To preview our results, we found that a single class of models, namely scaled evi-dence strength models, provided the best account of confidence across all task-modality con-figurations. However, while a single class of models provided the best account across modalities and tasks, we found it is unlikely that the metacognitive process operates under the same parameter settings across modalities. We tested several levels of flexibility in the parame-ter settings of a model fit to data from both modalities and found that the best model was one in which all the parameters were estimated separately in each modality. We propose that a scaled evidence strength algorithm tuned to the specific information in each sensory modality provides the best account of metacognition across domains.

## Results

### Participant exclusion

To ensure that participants understood the task and the underlying category distributions, we excluded data from individuals whose categorisation accuracy was not significantly above

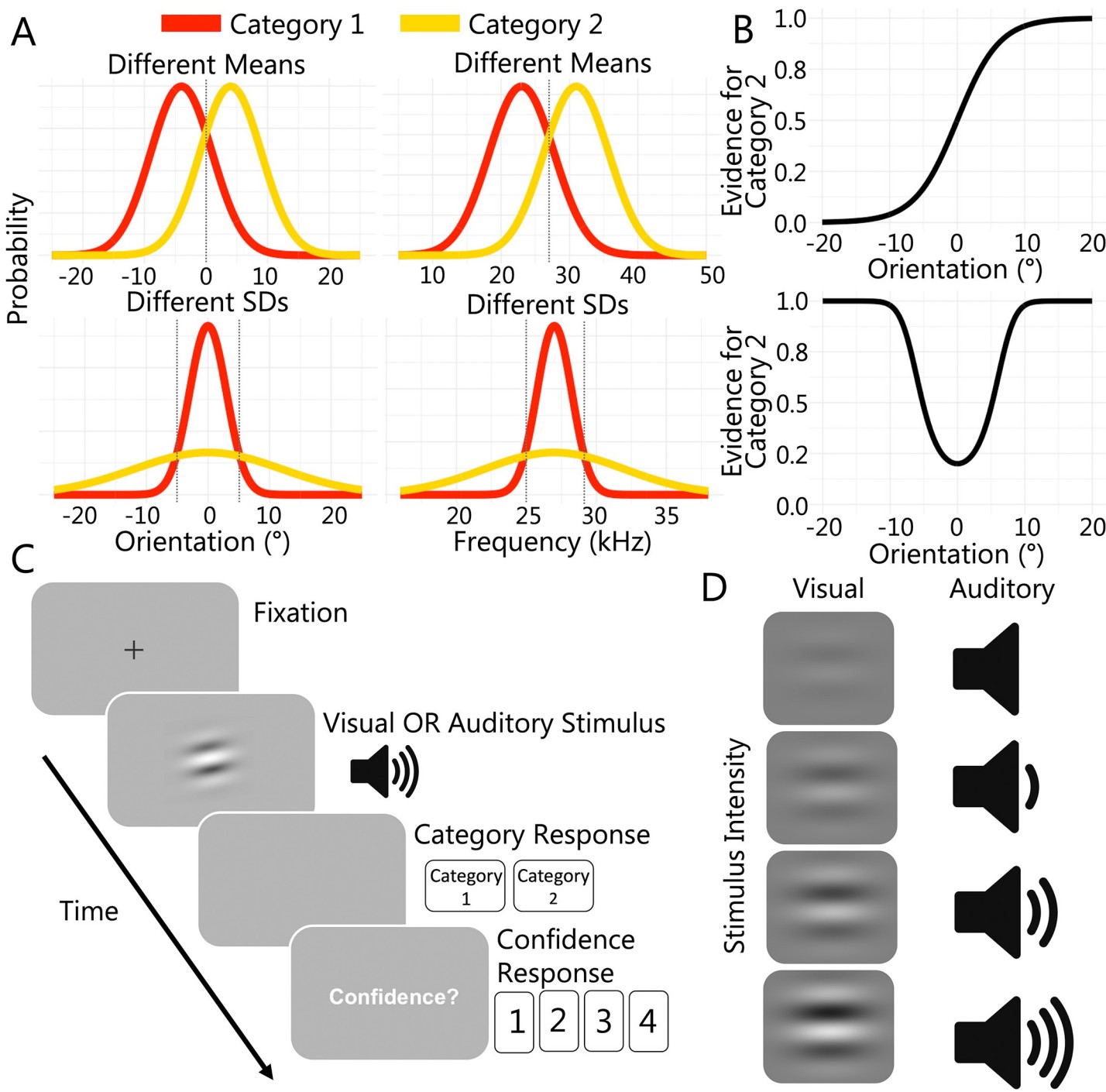

**Fig 2. Experimental Task.** (A) Category distributions for all task-modality configurations. (B) Changes in evidence for category 2 as a function of stimulus value in the different means task (top panel) and the different SDs task (bottom panel). (C) Trial sequence for test trials. (D) Levels of sensory uncertainty across modalities, where Gabor contrast varied in the visual tasks and tone intensity varied in the auditory tasks.

chance (50%) in the highest intensity condition, using a binomial test of significance. This test was applied to each task and modality separately and included testing trials only (see **S1 Table**). We considered a category response as correct if it was the most probable generating

category, given the stimulus value and category distributions for that task. One of the 12 participants was excluded using this criterion. A second participant was excluded for reported confusion about the task instructions, a third was excluded for not completing all sessions (2/4 sessions) and a fourth was excluded for using the highest confidence rating on every trial in one session. The target sample size (after exclusions; $N = 8$) was preregistered and was determined a priori (https://osf.io/d3y5n) because it allowed full counterbalancing of experimental conditions.

## Confirming the effect of the chosen stimulus manipulations: Model-free GLMM

Although our primary theoretical interest was in the model-based analysis of the data, we first checked that our chosen stimulus manipulations had the intended effect on category and confidence responses across modalities. As described below, we fit generalised linear mixed models (GLMMs) to predict category and confidence responses from the relevant stimulus features for each modality and task. In **Fig 3A**, we show the raw data (without GLMM predictions) for each modality and task (columns). Stimulus values, orientation for the visual tasks and frequency for the auditory tasks, are binned on the x axis so that each bin contains the same number of trials. Combined category and confidence responses, on the y axis, are coded such that value indicates confidence (1, 2, 3, 4) and polarity indicates category, where category 1 responses are negative and category 2 responses are positive.

Because the expected and observed pattern of responses differed across the different means and different SDs task (see **Fig 2B**), we had to transform the stimulus values to fit the GLMMs and compare the results across the different versions of the task (as in **Fig 3B** and **3C**).

**Confirming that the stimulus manipulations affected category responses.**   To determine the effect of the stimulus manipulations on category responses, we transformed stimulus values to 'evidence for category 2'. For each task and modality, we computed the evidence for category 2 by evaluating the probability density of the presented stimulus value for the category 2 distribution, normalised by the total probability density for both category distributions for that stimulus value,

$$E(C = 2) = \frac{N(s; \mu_2, \sigma_2)}{\sum_{C=1}^{2} N(s; \mu_C, \sigma_C)} \tag{1}$$

where $s$ is the true value of the stimulus, $C$ is the category, $\mu_2$ and $\sigma_2$ are the mean and standard deviation of the category 2 distribution and $N$ denotes the normal density function.

Eq 1 computes evidence for category 2 for each stimulus value, which could range from 0 (stimulus could be from category 1 only) to 1 (stimulus could be from category 2 only). Note, however, that for the different SDs task evidence for category 2 never reaches 0 because the category 1 distribution falls entirely within the range of the category 2 distribution. The minimum evidence value for this task is 0.2 (see **Fig 2B**).

We fit linear mixed models using a logistic link function to predict category response (1 or 2) from stimulus intensity and evidence for category 2, for each task and modality. The models included an intercept term, intensity, evidence and the interaction term between intensity and evidence as random effects. For each regression analysis, we report the odds ratio for the fixed effect and 95% confidence intervals in **Table 1**. As this analysis was intended to provide a descriptive summary of the effect of the stimulus manipulations on responses across modalities, we report other statistical metrics and follow-ups in **S2 Table**.

As expected, stimulus values that indicated increasing evidence for category 2 were positively predictive of category 2 responses in all tasks and modalities (see 'Evidence for Category

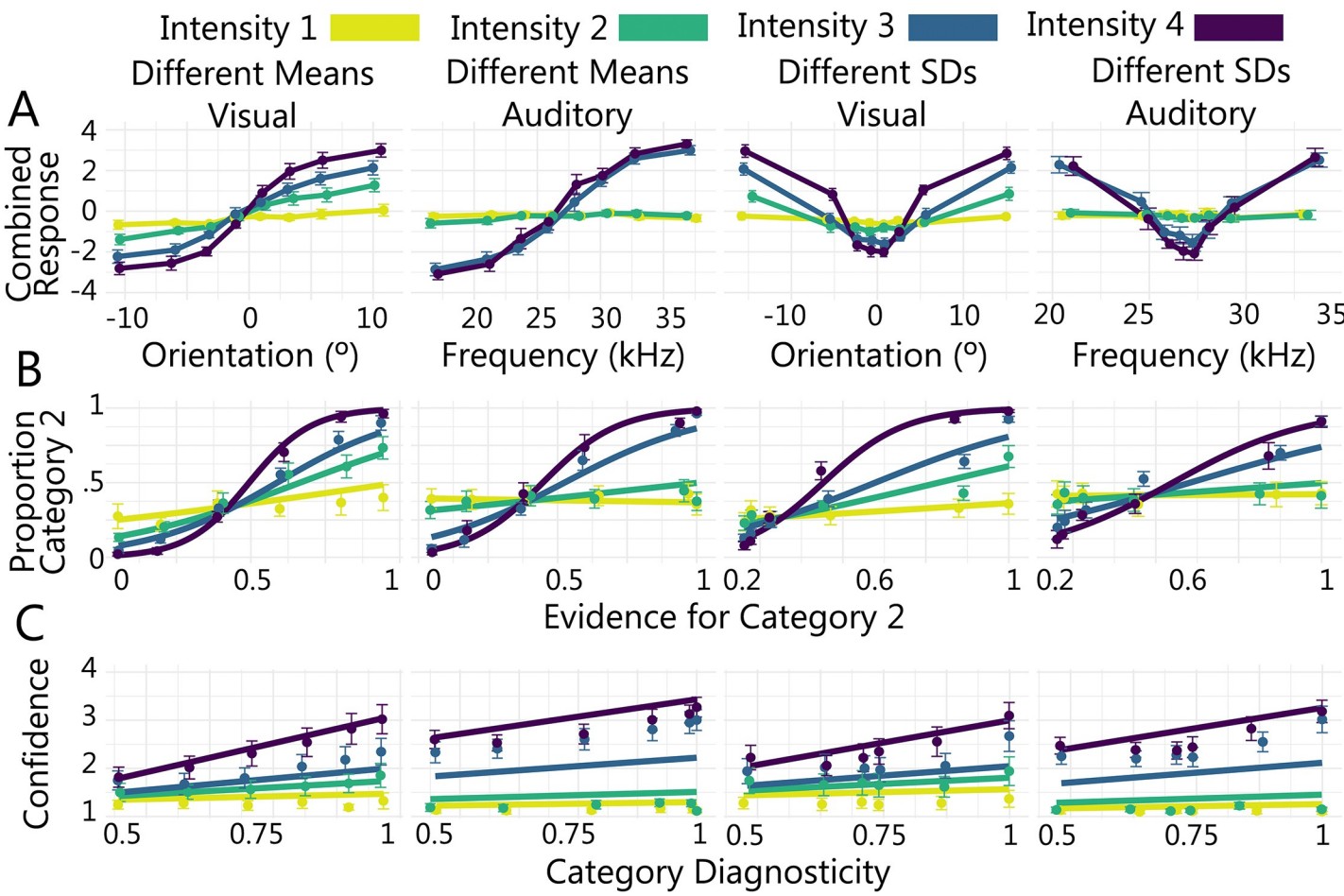

**Fig 3. Logistic and Linear Mixed Models.** (A) Mean combined category and confidence response across all participants, where value indicates confidence and polarity indicates category, as a function of binned stimulus value. (B) Model-free GLMM. Category decisions as a function of category 2 evidence. Data points show empirical data. Solid lines show logistic model predictions for the fixed effect of evidence at each intensity level. (C) Model-free GLMM. Confidence as a function of category diagnosticity. Data points show empirical data. Solid lines show linear model predictions for the fixed effect of category diagnosticity at each intensity level. Error bars show ± 1 *SEM*. Where error bars are not visible, they are occluded by the plotting symbols.

2' in top panel of **Table 1** for odds ratios). Increasing stimulus intensity was also positively predictive of category 2 responses for all tasks except the auditory different SDs task (see 'Stimulus Intensity' in top panel of **Table 1** for odds ratios). These main effects were qualified by significant two-way interactions between category 2 evidence and intensity, for all tasks and modalities (see 'Evidence × Intensity Interaction' in top panel of **Table 1** for odds ratios). As illustrated in **Fig 3B**, the effect of increasing category 2 evidence on category 2 responses was strongest for the highest intensity stimuli and decreased with stimulus intensity (see **S2 Table** for follow up tests). Of note, in the auditory modality the empirical data cluster across intensity levels, suggesting that participants did not appear to be able to distinguish between the two lowest intensity levels and the two highest intensity levels in the auditory modality in the same way as they did in the visual modality. The model predictions, however, do not capture this effect directly. We provide additional discussion of this feature of the data in **S1 Text** (see **Figs A and B and Tables A and B**).

These findings demonstrate that participants understood the category distributions across tasks and modalities. Specifically, when there was an increase in the relative probability that a

**Table 1.** *GLMM with Stimulus Manipulations as Predictors.*

| GLMM Predictor | | Task-modality configurations | | | |
| --- | --- | --- | --- | --- | --- |
| | | Visual Different Means | Auditory Different Means | Visual Different SDs | Auditory Different SDs |
| | | Category | | | |
| | (Intercept) | 0.73 | 0.97 | 0.76* | 0.74 |
| Intercept | $CI_{lower}$ | 0.48 | 0.65 | 0.61 | 0.50 |
| | $CI_{upper}$ | 1.12 | 1.43 | 0.95 | 1.10 |
| Evidence for Category 2 | Odds Ratio | 6.37*** | 4.13*** | 3.81*** | 2.38*** |
| | $CI_{lower}$ | 4.22 | 3.06 | 3.01 | 1.71 |
| | $CI_{upper}$ | 9.61 | 5.58 | 4.81 | 3.32 |
| Stimulus Intensity | Odds Ratio | 1.42*** | 1.85* | 1.99*** | 1.24 |
| | $CI_{lower}$ | 1.16 | 1.08 | 1.49 | 0.73 |
| | $CI_{upper}$ | 1.75 | 3.17 | 2.66 | 2.10 |
| Evidence × Intensity Interaction | Odds Ratio | 4.27*** | 4.42*** | 3.10*** | 2.29*** |
| | $CI_{lower}$ | 2.78 | 3.03 | 2.60 | 1.66 |
| | $CI_{upper}$ | 6.56 | 6.44 | 3.70 | 3.16 |
| | | Confidence | | | |
| Intercept | (Intercept) | 1.82*** | 1.98*** | 1.88*** | 1.82*** |
| | $CI_{lower}$ | 1.45 | 1.82 | 1.46 | 1.66 |
| | $CI_{upper}$ | 2.18 | 2.14 | 2.30 | 1.99 |
| Category Diagnosticity | Estimate | 0.19*** | 0.13*** | 0.15*** | 0.13*** |
| | $CI_{lower}$ | 0.13 | 0.08 | 0.11 | 0.07 |
| | $CI_{upper}$ | 0.25 | 0.18 | 0.18 | 0.19 |
| Stimulus Intensity | Estimate | 0.40*** | 0.72*** | 0.38*** | 0.62*** |
| | $CI_{lower}$ | 0.26 | 0.54 | 0.26 | 0.45 |
| | $CI_{upper}$ | 0.55 | 0.91 | 0.51 | 0.80 |
| Diagnosticity × Intensity Interaction | Estimate | 0.15*** | 0.11*** | 0.10*** | 0.10*** |
| | $CI_{lower}$ | 0.10 | 0.06 | 0.07 | 0.05 |
| | $CI_{upper}$ | 0.21 | 0.15 | 0.13 | 0.15 |

*Note*. CI terms refer to 95% confidence intervals calculated using the profile likelihood method. Significance values were obtained using the Satterthwaite approximation to calculate the degrees of freedom for the t-distribution based on the estimated variance-covariance matrix of the model parameters [47]. For category GLMMs (top section), we report the odds ratio (exponentiated coefficient estimate) for ease of interpretation. The odds ratio represents the change in odds of the outcome for a one-unit change in the predictor variable. For confidence GLMMs (bottom section), we report standardised regression coefficients.

*$p < 0.050$

**$p < 0.010$

***$p < 0.001$

stimulus was sampled from category 2, participants were more likely to report that that stimulus had been sampled from category 2. This was true regardless of the specific category structure (i.e., for both the different means and different SDs tasks) and stimulus modality. The effect of category 2 evidence was modulated by stimulus intensity, contrast for the visual tasks and loudness for the auditory tasks, such that as the intensity of the stimulus was reduced, category responses were more random and less influenced by the category distributions. This interaction between evidence and intensity was observed across modalities and confirmed that the chosen stimulus manipulations had the intended effect on participants' category responses in both the visual and auditory domains. Because we were interested in quantifying the computations underlying metacognitive judgements across modalities, this analysis also provided

evidence for similarities in the perceptual task requirements, allowing us to isolate any differences in metacognition itself [46].

**Confirming that the stimulus manipulations affected confidence.** As above, the expected pattern of confidence responses differed across tasks (see **Fig 2B**). To compare the effect of the stimulus manipulations on confidence across the different versions of the tasks, we transformed stimulus values to 'category diagnosticity' prior to fitting the GLMMs. For each task and modality, we computed category diagnosticity by evaluating the probability density of the presented stimulus value for the most probable category distribution, normalised by the total probability density for both category distributions:

$$CD = \frac{\max\left(N(s; \mu_1, \sigma_1), N(s; \mu_2, \sigma_2)\right)}{\sum_{C=1}^{2} N(s; \mu_C, \sigma_C)} \tag{2}$$

where $s$ is the true value of the stimulus, $C$ is the category, $\mu_1$ and $\sigma_1$ are the mean and standard deviation of the category 1 distribution, $\mu_2$ and $\sigma_2$ are the mean and standard deviation of the category 2 distribution and $N$ denotes the normal density function.

Eq 2 computes category diagnosticity for each stimulus value, which could range from 0.5 (both categories are equally likely) to 1 (stimulus could be from the most probable category only). We fit linear mixed models to predict confidence from stimulus intensity and category diagnosticity, for each task and modality. The models included an intercept term, intensity, diagnosticity and the interaction term between intensity and diagnosticity as random effects. For each regression analysis, we report the fixed effect and 95% confidence intervals in **Table 1**.

Increasing category diagnosticity (see 'Category Diagnosticity' in bottom panel of **Table 1** for coefficients) and stimulus intensity (see 'Stimulus Intensity' in bottom panel of **Table 1** for coefficients) were predictive of increasing confidence in all tasks. These main effects were qualified by significant two-way interactions between category diagnosticity and intensity for all tasks (see 'Diagnosticity × Intensity Interaction' in bottom panel of **Table 1** for coefficients). Increasing category diagnosticity was associated with increasing confidence but the strength of this effect was modulated by stimulus intensity. As illustrated in **Fig 3C**, the effect of increasing category diagnosticity was strongest for the highest intensity stimuli and decreased with lower stimulus intensities (see **S2 Table** for follow up tests). As observed in the category response data, the empirical confidence data also cluster across intensity levels in the auditory modality. See **S1 Text** for additional discussion.

Overall, these results indicate that as the relative evidence for a given category increased, participants' confidence in their chosen category increased. Taken with the results of the category GLMMs, showing that participants' category responses corresponded with the most probable category given the value of the stimulus, these findings suggest that participants understood the category distributions and used them to accurately guide their confidence responses. The effect of the category distributions on confidence was modulated by stimulus intensity, such that as the intensity of the stimulus decreased, confidence responses were uniformly low. This effect was observed across both modalities.

To further investigate similarities in the category and confidence response profiles across modalities, we also fit GLMMs to the evidence and intensity data from all tasks and modalities concurrently (see **S2 Text**). This allowed us to investigate if all the data could be described by the same set of effects or alternatively, if there were interactions between the effect of certain stimulus manipulations and the different task-modality configurations. These analyses (see **Table A** and **Fig A** in **S2 Text**) provided converging evidence that participants' category and

confidence judgements were associated with the stimulus features of interest in a similar way across modalities and tasks.

## Computational modelling

The GLMMs showed that our chosen stimulus manipulations had the intended effects on category and confidence responses across modalities. We next sought to characterise the processes underlying the category and confidence judgements using computational modelling. We evaluated the performance of popular confidence models from three classes: 1) unscaled evidence strength models, 2) scaled evidence strength models and 3) Bayesian models (see **Fig 4**). To assess the similarity in the underlying computational processes, we sought to determine whether a single class of models provided the best account of responses across modalities and tasks (visual different means, auditory different means, visual different SDs, auditory different

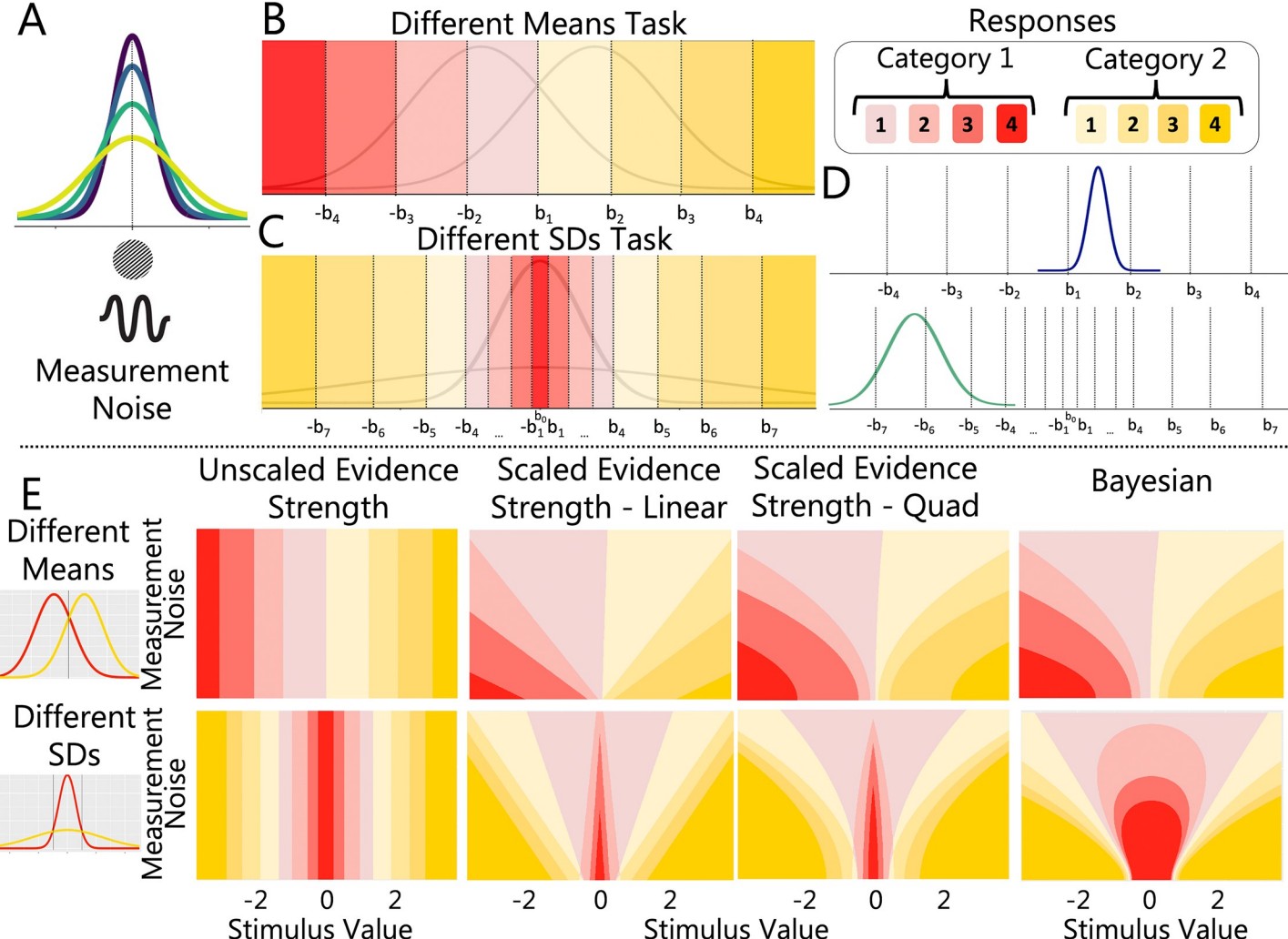

**Fig 4. Computational Models.** (A) Pictorial representation of the measurement distribution showing how measurement noise varies as a function of stimulus intensity. Illustration of boundary placement and response regions for (B) the different means task and (C) the different SDs task. The colours represent different category responses and saturation represents associated confidence ratings. (D) The measurement distribution is compared to the boundaries. (E) Predicted response regions for the different means (top panel) and different SDs (bottom panel) tasks across different levels of sensory uncertainty and standardised stimulus values for the different models. These predictions were generated using parameter values similar to those found after fitting participant data.

SDs). An overview of the computational mechanisms in each class and the performance of each model is provided in detail below (see **S3 Text** for alternative visualisations of model fits).

### Unscaled evidence strength models: The distance model

According to unscaled evidence strength models, confidence reports depend on the distance between the sensory measurement and the category decision criterion. These models do not independently quantify sensory uncertainty. We tested a single variation of this class of models: the distance model. In the distance model, the observer compares their internal representation of a stimulus to a set of category/confidence boundaries that do not change with stimulus intensity.

To visualise the fit of the distance model, in **Fig 5** we show the model's predicted responses (solid lines) against the empirical data (square plotting symbols) as a function of standardised stimulus values (see **Fig A** in **S3 Text** for alternative visualisations of model fits and **Standardisation of Stimulus Values** for stimulus standardisation procedure). The empirical data showed a clear effect of stimulus intensity (denoted by different colours) on category and confidence responses. The distance model, however, had no mechanism to modulate responses by

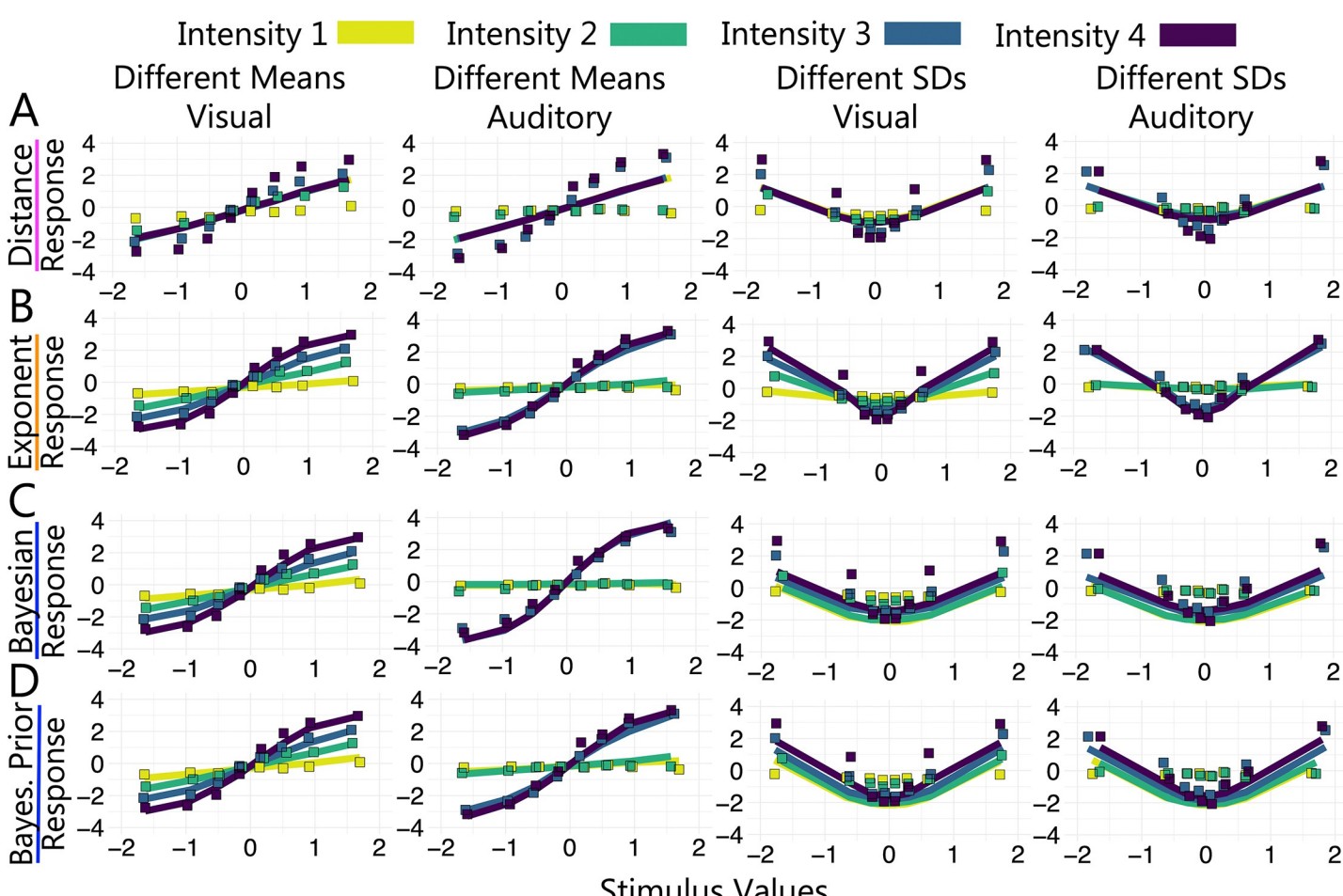

**Fig 5. Model Predictions and Experimental Data.** Columns show different versions of the task. Square plotting symbols show empirical data and solid lines show model predictions. Rows show model predictions for (A) the distance model (unscaled evidence strength class; pink), (B) the free-exponent model (scaled evidence strength class; orange), (C) the log posterior probability ratio model (Bayesian class; blue) and (D) the log posterior probability ratio model with free category distribution parameters (Bayesian class; blue).

stimulus intensity. Unlike scaled evidence strength models, the boundary positions were not dependent on the estimate of measurement noise for a given intensity, so the model was forced to settle on a set of boundaries that best captured responses across all intensities. As a result, the model overpredicted confidence for low intensities and underpredicted confidence for high intensities. This pattern of over- and under-prediction was seen across modalities and tasks. Overall, the distance model did not perform well, suggesting that a model in which confidence depends entirely on the distance from the category decision criterion provides a poor account of observed responses in both the visual and auditory tasks.

### Scaled evidence strength models: Linear, quadratic & free-exponent models

According to scaled evidence strength models, confidence reports depend on the distance between the observer's internal representation of the stimulus and sensory uncertainty dependent choice and confidence boundaries. We tested three variations of this class of models which used different scaling rules: the linear model, the quadratic model and the free-exponent model.

Each model allowed for a point estimate of measurement noise to expand/contract the category/confidence boundaries and thereby account for the changes in responding across stimulus intensities seen in the empirical data. We show the predictions for the free-exponent model in **Fig 5B**. See **Fig B** in **S3 Text** for linear model predictions and **Fig C** in **S3 Text** for quadratic model predictions. All members of this model class fit the data well and the differences in the predictions across models are subtle.

The additional free parameter in the free-exponent model allowed for the changes in boundary positions across intensities to occupy values between and beyond linearly- or quadratically-spaced values. Thus, the free-exponent model could account for a wider range of sensory uncertainty effects across participants. This additional flexibility was best able to capture the empirical data on average and thus, the model had the lowest summed *AIC* and *BIC* scores across tasks and modalities (with the exception of summed *BIC* for the linear model in the auditory different means task and quadratic model in the visual different SDs task; see **Table 2**). See **Table 3** for means and standard deviations for boundary and σ parameters.

As shown in **Table 2**, adding an orientation-dependent noise parameter for modelling performance in the visual modality provided a marginal improvement in the performance of the free-exponent model in the different means task and the different SDs task. Including the orientation-dependent noise parameter, however, did not qualitatively change the visualisation of the model predictions (see **Fig D** in **S3 Text**) or the conclusions (that is, which class of model was the best performing across modalities and tasks).

Overall, the scaled evidence strength models were on average the best performing models across tasks and modalities, see **Fig 6**. This finding suggests that a model in which participants use sensory uncertainty dependent choice and confidence boundaries provides a good account of observed responses in both the visual and auditory tasks.

### Bayesian models

According to Bayesian models of confidence, observers optimally combine sensory information with prior knowledge of the generative model to form a posterior belief about the nature of the stimulus. We tested several variants of this class of models.

The Bayesian models had fewer parameters than the scaled evidence strength models because the boundaries had the same positions in probability space across intensity levels. The estimates of measurement noise at each intensity level allowed the boundaries to shift in perceptual space. As shown in **Fig 5C and 5D**, this allowed the model to account for different

**Table 2.** *Model Fits.*

| Model Class | Model | Visual | | | Auditory | | |
|---|---|---|---|---|---|---|---|
| | | $AIC_{sum}$ | $BIC_{sum}$ | Preferred Model[a] | $AIC_{sum}$ | $BIC_{sum}$ | Preferred Model[a] |
| | | | | Different Means Task | | | |
| Unscaled ES | Distance | 16666.53 | 17092.68 | 0 | 18909.84 | 19336.00 | 0 |
| Scaled ES | Linear | 14189.84 | 14829.07 | 0 | 13029.62 | **13668.85** | **5** |
| Scaled ES | Quadratic | 14196.21 | 14835.44 | 2 | 13052.21 | 13691.44 | 2 |
| Scaled ES | Free-Exponent | 14148.15 | 14840.65 | 0 | **13020.72** | 13713.22 | 1 |
| Scaled ES | Free-Exponent + ODN | **13802.79** | **14744.56** | 6 | - | - | - |
| Bayesian | LPPR | 14430.08 | 14856.24 | 0 | 14291.57 | 14717.73 | 0 |
| Bayesian | LPPR + ODN | 14525.76 | 15005.18 | 0 | - | - | - |
| Bayesian | LPPR + D noise | 14421.41 | 14900.83 | 0 | 14323.07 | 14802.49 | 0 |
| Bayesian | LPPR: Free Prior | 14422.19 | 14954.88 | 0 | 14206.87 | 14739.56 | 0 |
| | | | | Different SDs Task | | | |
| Unscaled ES | Distance | 17146.46 | 17732.42 | 0 | 18398.23 | 18984.19 | 0 |
| Scaled ES | Linear | 14658.60 | 15617.45 | 2 | 13284.32 | 14243.17 | 3 |
| Scaled ES | Quadratic | 14598.85 | **15557.71** | **4** | 13256.32 | 14215.17 | 0 |
| Scaled ES | Free-Exponent | 14669.23 | 15681.35 | 0 | **13190.29** | **14202.42** | **5** |
| Scaled ES | Free-Exponent + ODN | **14541.42** | 15886.81 | 2 | - | - | - |
| Bayesian | LPPR | 16869.92 | 17455.89 | 0 | 18392.48 | 18978.44 | 0 |
| Bayesian | LPPR + ODN | 18711.50 | 19350.74 | 0 | - | - | - |
| Bayesian | LPPR + D noise | 16839.16 | 17478.39 | 0 | 17746.37 | 18385.61 | 0 |
| Bayesian | LPPR: Free Prior | 16914.33 | 17606.84 | 0 | 18487.76 | 19180.26 | 0 |

*Note.* ES in model class labels refers to evidence strength. LPPR refers to the log posterior probability ratio models, ODN refers to models which included an orientation dependent noise parameter and D noise refers to models with a decision noise parameter. [a]The number of participants for whom the model was the best fitting model. Where there was inconsistency between *AIC* and *BIC* for the best fitting model, we chose the model with the lowest *AIC*, based on the results of the model recovery. Bolded values indicate best performing model according to a given metric.

patterns of responding across intensities. This mechanism appeared sufficient for the different means task where overall the Bayesian models fit the data well in both modalities—though not as well as the scaled evidence strength models (see **Fig 5**). In the different SDs task, however, the Bayesian models produced qualitative mispredictions of category and confidence data. We discuss the performance of each model in the Bayesian class in turn below (with emphasis on its performance in the different SDs task).

**Log Posterior Probability Ratio.** In the log posterior probability ratio model, we assumed that the observer compared the log posterior probability ratio of the stimulus measurement to a set of category and confidence boundaries in log posterior probability ratio space. In the different SDs task, the main failure of this model was that it overpredicted confidence at lower intensity levels across modalities (see **Fig E** in **S3 Text**). Even though the boundary positions were sensitive to the estimates of $\Sigma$ and the model was able to generate different predictions at each intensity level, the boundary parameters themselves were shared across all intensity levels such that the fit to low-intensity trials was constrained by high-intensity trials. This constraint did not appear to allow sufficient flexibility in the model to account for the distinct patterns of responses across intensity levels. We tested several variations of the log posterior probability ratio model which relaxed the assumptions of the Bayesian framework.

**Log Posterior Probability Ratio with Orientation Dependent Noise.** Adding an orientation-dependent noise parameter did not improve the fit of the log posterior probability ratio model to visual modality data in either the different means task or the different SDs task. The

**Table 3. Means and SDs for Best Fitting Noise and Boundary Parameters for Free-Exponent Model.**

| Parameter | Intensity 1 | | Intensity 2 | | Intensity 3 | | Intensity 4 | |
|---|---|---|---|---|---|---|---|---|
| | *M* | *SD* | *M* | *SD* | *M* | *SD* | *M* | *SD* |
| | Visual Different Means | | | | | | | |
| $\sigma$ | 5.36 | 2.33 | 1.72 | 0.44 | 1.04 | 0.33 | 0.66 | 0.23 |
| $b_1$ | 3.11 | 10.32 | 0.63 | 3.97 | 0.24 | 2.52 | 0.05 | 1.71 |
| $b_2$ | 9.10 | 8.84 | 2.25 | 3.62 | 1.09 | 2.12 | 0.54 | 1.15 |
| $b_3$ | 16.76 | 5.20 | 4.38 | 2.08 | 2.25 | 1.28 | 1.23 | 0.72 |
| $b_4$ | 23.31 | 4.14 | 6.39 | 0.90 | 3.60 | 0.40 | 2.35 | 0.12 |
| | Auditory Different Means | | | | | | | |
| $\sigma$ | 7.70 | 2.72 | 5.47 | 2.52 | 0.89 | 0.38 | 0.77 | 0.30 |
| $b_1$ | 1.92 | 2.76 | 1.09 | 1.64 | 0.01 | 0.03 | -0.02 | 0.07 |
| $b_2$ | 14.31 | 4.63 | 8.64 | 5.15 | 0.19 | 0.24 | 0.08 | 0.14 |
| $b_3$ | 19.99 | 6.48 | 12.12 | 5.25 | 0.92 | 0.62 | 0.72 | 0.46 |
| $b_4$ | 26.76 | 6.53 | 16.93 | 6.79 | 2.02 | 1.33 | 1.71 | 0.99 |
| | Visual Different SDs | | | | | | | |
| $\sigma$ | 2.32 | 1.17 | 1.20 | 0.29 | 0.82 | 0.26 | 0.57 | 0.25 |
| $b_1$ | 0.04 | 0.09 | 0.08 | 0.13 | 0.08 | 0.13 | 0.09 | 0.13 |
| $b_2$ | 0.11 | 0.18 | 0.17 | 0.17 | 0.18 | 0.16 | 0.20 | 0.16 |
| $b_3$ | 0.55 | 0.63 | 0.43 | 0.26 | 0.48 | 0.19 | 0.51 | 0.16 |
| $b_4$ | 2.90 | 1.99 | 1.34 | 0.47 | 0.94 | 0.34 | 0.79 | 0.24 |
| $b_5$ | 4.93 | 1.70 | 2.53 | 1.45 | 1.54 | 1.09 | 1.09 | 0.70 |
| $b_6$ | 7.18 | 1.90 | 3.78 | 2.56 | 2.30 | 1.95 | 1.64 | 1.34 |
| $b_7$ | 11.87 | 6.91 | 7.61 | 7.69 | 4.73 | 4.80 | 4.20 | 4.66 |
| | Auditory Different SDs | | | | | | | |
| $\sigma$ | 2.26 | 0.66 | 2.20 | 0.64 | 0.85 | 0.45 | 0.70 | 0.30 |
| $b_1$ | 0.003 | 0.01 | 0.01 | 0.01 | 0.07 | 0.09 | 0.07 | 0.09 |
| $b_2$ | 0.02 | 0.02 | 0.03 | 0.03 | 0.23 | 0.22 | 0.25 | 0.23 |
| $b_3$ | 0.24 | 0.34 | 0.28 | 0.34 | 0.79 | 0.14 | 0.81 | 0.16 |
| $b_4$ | 2.43 | 1.91 | 2.38 | 1.93 | 0.93 | 0.22 | 0.88 | 0.18 |
| $b_5$ | 5.88 | 2.87 | 5.57 | 2.68 | 1.15 | 0.46 | 0.97 | 0.23 |
| $b_6$ | 8.08 | 3.43 | 7.72 | 3.39 | 1.72 | 0.90 | 1.45 | 0.55 |
| $b_7$ | 13.16 | 8.05 | 12.78 | 8.25 | 3.11 | 2.46 | 2.25 | 0.98 |

*Note*. Means and standard deviations were calculated using best fitting parameters for individual participants. $b_1$-$b_4$ were the boundary parameters for the different means tasks and $b_1$-$b_7$ were the boundary parameters for the different SDs tasks. Boundary parameters were calculated using $k+m\sigma^a$ for each participant (see Free-Exponent model for further description and **Best Fitting Model Parameters** for details of all model parameters).

predictions of this model still showed the same systematic deviations from the data as the log posterior probability ratio model. Specifically, both models overpredicted confidence at lower intensity levels across tasks (see **Fig F** in **S3 Text**).

**Log Posterior Probability Ratio with Decision Noise.** Adding a Gaussian noise term on the calculation of the log posterior probability ratio improved the fit of the log posterior probability ratio model in all tasks and modalities, except the auditory different means configuration. Including the decision noise parameter shifted the best-fitting boundary parameters away from the means of the category distributions. As a result, the model predicted more low confidence, category 1 responses at the lower intensity levels for all stimulus values (see **Fig G** in **S3 Text**). This aligned with the data better in terms of confidence but overpredicted category 1 responses as the experimental data showed roughly equal proportions of category 1 and

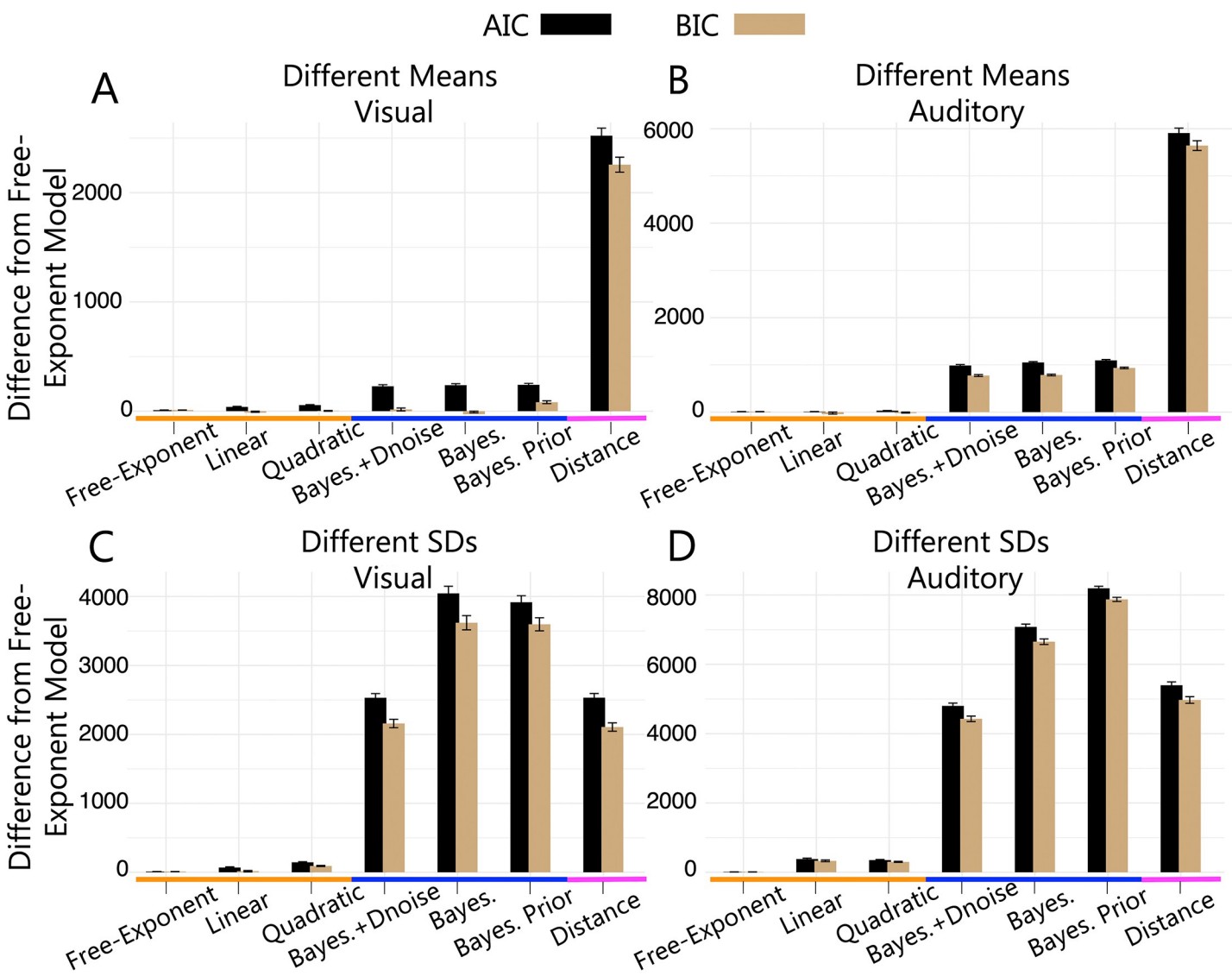

**Fig 6. Model Comparison.** Difference in summed *AIC* (black) and *BIC* (brown) between the free-exponent model and other models from 3 classes: unscaled evidence strength model (pink; the distance model), scaled evidence strength models (orange; free-exponent, linear and quadratic models) and Bayesian models (blue; the log posterior probability ratio model labelled as 'Bayes.'; the log posterior probability ratio model with decision noise labelled as 'Bayes.+Dnoise'; and the log posterior probability ratio model with free category distribution parameters labelled as 'Bayes. Prior'). (A) Different means visual task, (B) different means auditory task, (C) different SDs visual task and (D) different SDs auditory task.

category 2 responses for a wide range of stimulus values at low intensities (i.e., participants are likely *guessing* a category more often). As a result of the shift in boundary positions away from the centre of the category distributions across all intensity levels, the model underpredicted confidence for stimulus values close to the mean of the category distributions and also under-predicted category 2 responses for extreme stimulus values at the higher intensity levels. In the different SDs task, the log posterior probability ratio model with decision noise was the best performing of the Bayesian models in both modalities. In the different means task, it was also the best performing Bayesian model in the visual modality. In the auditory modality, the log posterior probability ratio model with decision noise was outperformed by the log posterior probability ratio model with free category distribution parameters see Table 2.

**Log Posterior Probability Ratio with Free Category Distribution Parameters.** The log posterior probability ratio model assumed that the observer knew the true values of the means and standard deviations of the category distributions. We tested a model in which the observer had imperfect knowledge of the parameters of the generative model, and we estimated these values as free parameters instead. Like the standard log posterior probability ratio model, this model overpredicted confidence at lower intensities and underpredicted confidence for stimulus values at the centre of the distributions at the highest intensity (see **Fig H** in **S3 Text**). This misprediction was seen across modalities. The model did, however, do a better job of matching confidence and proportion of category 2 responses at the more extreme values at higher intensities (relative to the log posterior probability ratio model with and without decision noise).

In the different means task, the best-fitting category distribution parameters were similar to the true category distribution parameters (see **Best Fitting Model Parameters**). In the different SDs task, the best-fitting category distribution parameters had a smaller standard deviation for the category 1 distribution and a larger standard deviation for the category 2 distribution in both modalities (see **Best Fitting Model Parameters**). The performance of this model demonstrated that even with a data-informed subjective prior, the Bayesian model class still produced qualitative mispredictions of category and confidence data in both the visual and auditory tasks. Overall, it appeared that regardless of the flexibility of the model, observed responses in the visual and auditory tasks were not fully consistent with a Bayesian framework in either modality.

## Consistency across participants

**Consistency in Best Performing Model Class.** The analyses described so far characterised how well each model captured the general patterns of responding across participants. As models were fit to data from each individual, however, we were also able to examine the best-fitting model for each participant. If a single class of models reliably characterised the computational processes underlying participants' category and confidence decisions collectively, we would expect that data from individual participants should also be best characterised by the same class of models across modalities and tasks. If, however, a single class of models did not reliably characterise the underlying computational processes, we would see variation in the performance of the model classes across individuals. Our goal was not to investigate individual differences per se, which would require a larger sample size, but to examine the consistency in model fits across participants.

As shown in **Table 2**, in both the visual and auditory modality the scaled evidence strength models performed best for all participants. This analysis revealed that even at the individual level, there was a preference for the scaled evidence strength class across modalities and tasks, consistent with the summed *AIC* and *BIC* scores which were the lowest for the scaled evidence strength models, as described above.

**Consistency in Best Performing Model.** While the main motivation of the model-based comparison was to focus the analysis at the level of model classes, to further understand the specific implementation of the scaled evidence strength models, we also examined differences at the individual model level. We compared the best performing model for each participant and found that within the scaled evidence strength class, there were notable differences in model performance.

In the auditory different means task, either the linear model (62.5% of participants), quadratic model (25% of participants) or free-exponent model (12.5%) performed best for individuals, while the free-exponent model was the best performing model on average. For the visual different means task, by contrast, either the free-exponent model (with orientation dependent

noise; 75% of participants) or the quadratic model (25% of participants) was the best performing model for participants best characterised by a scaled evidence strength class model. In the different SDs task in the visual modality, either the linear (25%), quadratic (50%) or free-exponent (25%) performed best and in the auditory modality, the free-exponent model was preferred (62.5%; remaining 37.5% for linear). Therefore, although the scaled evidence strength class consistently outperformed the other model classes, a specific scaling rule within that class (linear, quadratic or other exponent) did not seem to apply consistently across different versions of the task, across modalities or across participants.

## Parameter settings across modalities

In the analyses described thus far we sought to determine whether a single class of metacognitive models could account for choice and confidence judgements across the visual and auditory modalities. Consistent with this, we found that the scaled evidence strength models were the best performing across both the different means and different SDs task in both modalities. We found consistent evidence for this class of models both at the group level, where the models were best able to capture the main patterns of responding across participants, and at the individual participant level, where data from most individual participants were best accounted for by the same class of models across tasks and modalities. Given the consistency in evidence for a single class of models, we next sought to determine the extent to which this scaled evidence strength algorithm required different parameter settings to accommodate responding across modalities.

If the scaled evidence strength algorithm operated in a "one size fits all" configuration and did not require tuning/parameterisation specific to each modality, a single set of parameters would be sufficient to account for category and confidence responses across modalities. Alternatively, if the algorithm was tuned/parameterised specifically in each modality, different sets of parameters would be required to account for responding across modalities. To address this question, rather than fit the model to each modality separately we fit the free-exponent model to data from both modalities simultaneously and formulated several levels of flexibility in the parameter settings of the free exponent model. To better visualise the fit of these models, in **Fig 7** we show model predictions (solid lines) and experimental data (square plotting symbols) as a function of standardised stimulus value (x axes) and confidence (y axes) for the least and most flexible models.

**Fig 7B** shows the common settings model, the least flexible of the models, in which a single set of parameters was fit to data from both modalities (see **Fig I** and **Fig J** in **S3 Text** for alternative visualisations of model predictions). The model predictions captured both the category and confidence data well. The common settings model, however, was outperformed by the more flexible models (see **Table 4**). The different noise settings model, where sigma was a free parameter for each intensity level in each modality, performed better than the common settings models. As shown in **Table 4**, the flexible settings model, where all parameters in the standard free-exponent model were freely estimated in each modality, was the best performing model in both the different means (see **Fig 7C**) and the different SDs tasks (see **Fig 7D**). This was true at the individual participant level and in terms of summed *AIC* and *BIC* for the group. These results show that adding flexibility in the parameter settings across modalities consistently improved the performance of the model (see **Fig 7C and 7D**). Overall, this finding suggests that although the scaled evidence strength algorithm seemed to provide the best account of confidence judgements across modalities, the process is tuned specifically in each modality and does not operate in a "one size fits all" configuration.

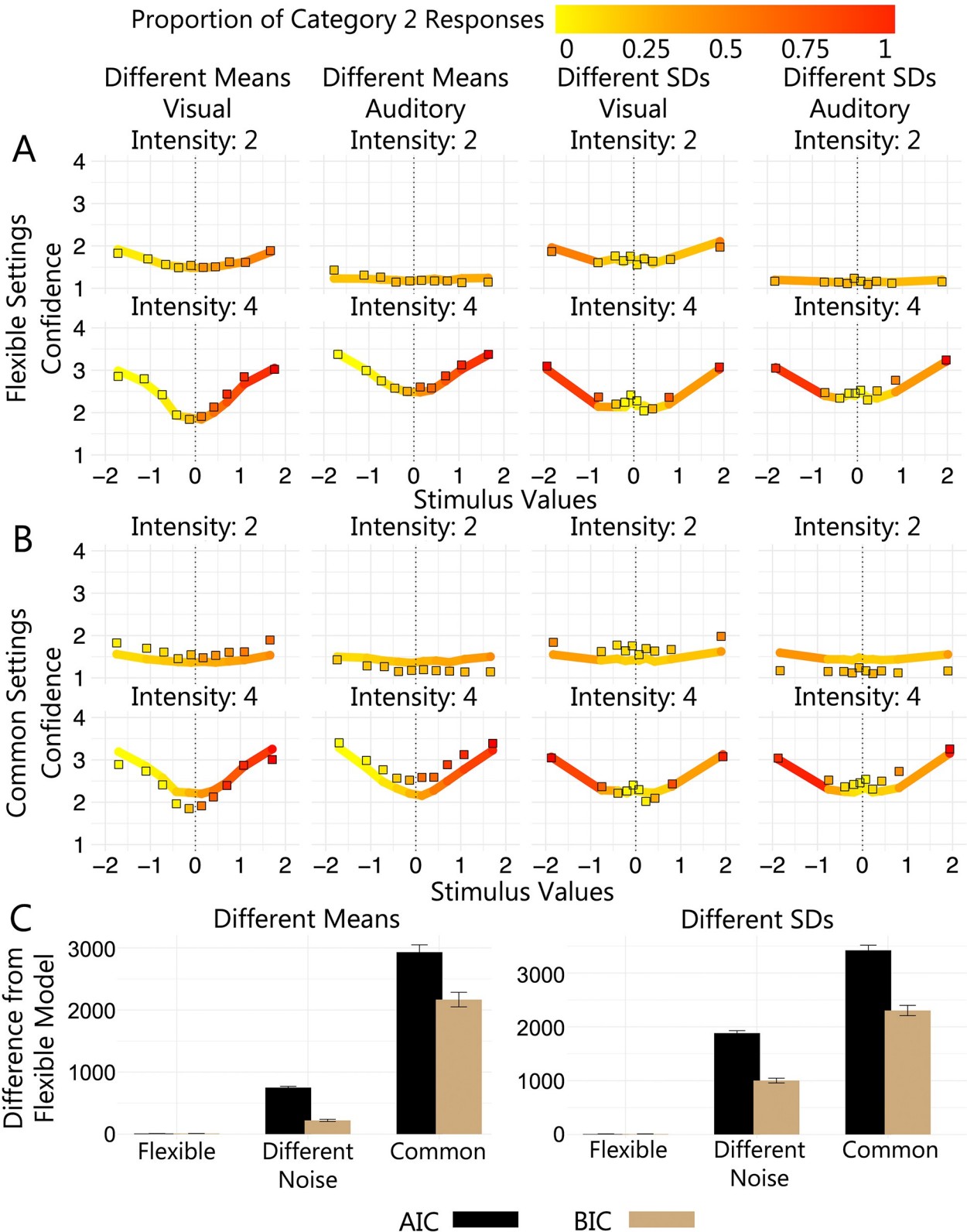

**Fig 7. Parameter Settings Across Modalities.** (A) and (B) show mean confidence and proportion of category 2 responses (colour scale) for binned stimulus values. Square plotting symbols show means for empirical data and solid lines show means for (A) flexible settings model predictions and (B) common settings model predictions. Columns show different versions of the task and rows show different intensity levels. For clarity the plots

include results for intensity levels 2 and 4 only. Difference in summed *AIC* (black) and *BIC* (brown) between the flexible settings model and the other models with different parameter settings in (C) the different means task and (D) different SDs task.

## Model recovery analysis

We also performed a model recovery analysis to test our ability to distinguish our core models and confirm our model-comparison results. For each model, we generated 100 synthetic datasets by simulating responses to a set of randomly sampled stimulus values that was similar to the empirical stimulus data. We simulated responses to these stimuli using randomly sampled parameter values that were similar to those obtained through fitting participant data. Each dataset contained 720 simulated 'trials', matching the size of the empirical dataset from each participant in the main experiment. We then fit all models to every dataset and calculated the probability that the model used to generate the data was the best fitting model, according to a given model selection metric, across all datasets. In **Fig 8A**, we show the model recovery results for a representative model from each class. We show the full details of model recovery using both *AIC* and *BIC* for model selection for all models in **Fig A** in **S4 Text**. As shown in **Fig 8A**, we found that across model classes, the simulated data were almost always best fit by the model that generated them. This result indicated that the model classes were distinguishable from one another and confirmed our model comparison results.

We also performed a model recovery analysis for the models used to evaluate the parameter settings across modalities. In **Fig 8B**, we show the results from these models, where we again found that the simulated data were almost always best fit by the model that generated them. See **Fig B** in **S4 Text** for full details of model recovery. This model recovery validated the model comparison results that were used to compare the parameter settings of the free-exponent model across the visual and auditory modalities (see **Fig 7**).

Overall, our models appeared to recover well. Importantly, we found that *AIC* scores produced better model recovery results, suggesting that for our models, the penalty for model complexity was too large in the calculation of *BIC*. We, therefore, used *AIC* scores for model selection where there were any differences in the best-fitting model during model comparison. We also report the parameter recovery properties of all models in **S5** and **S6 Text**s, showing that across different sample sizes the parameters recovered well for both the core models (see **Figs A—J** and **Tables A—D** in **S5 Text**) and for the models used to evaluate parameter settings across modalities (see **Figs A—C** and **Table A** in **S6 Text**).

**Table 4. Comparing Parameter Settings Assumptions Across Modalities.**

| Model | $AIC_{sum}$ | $BIC_{sum}$ | Preferred Model[a] |
|---|---|---|---|
| | Different Means Task | | |
| Common Settings | 30101.44 | 30866.03 | 0 |
| Different Noise | 27919.35 | 28919.20 | 0 |
| **Flexible Settings** | **27168.87** | **28698.05** | **8** |
| | Different SDs Task | | |
| Common Settings | 31280.86 | 32398.34 | 0 |
| Different Noise | 29743.64 | 31096.38 | 0 |
| **Flexible Settings** | **27859.53** | **30094.49** | **8** |

*Note*. Bolded values indicate best performing model according to a given metric.

[a]The number of participants for whom the model was the best fitting model. Where there was inconsistency between *AIC* and *BIC* for the best fitting model, we chose the model with the lowest *AIC*.

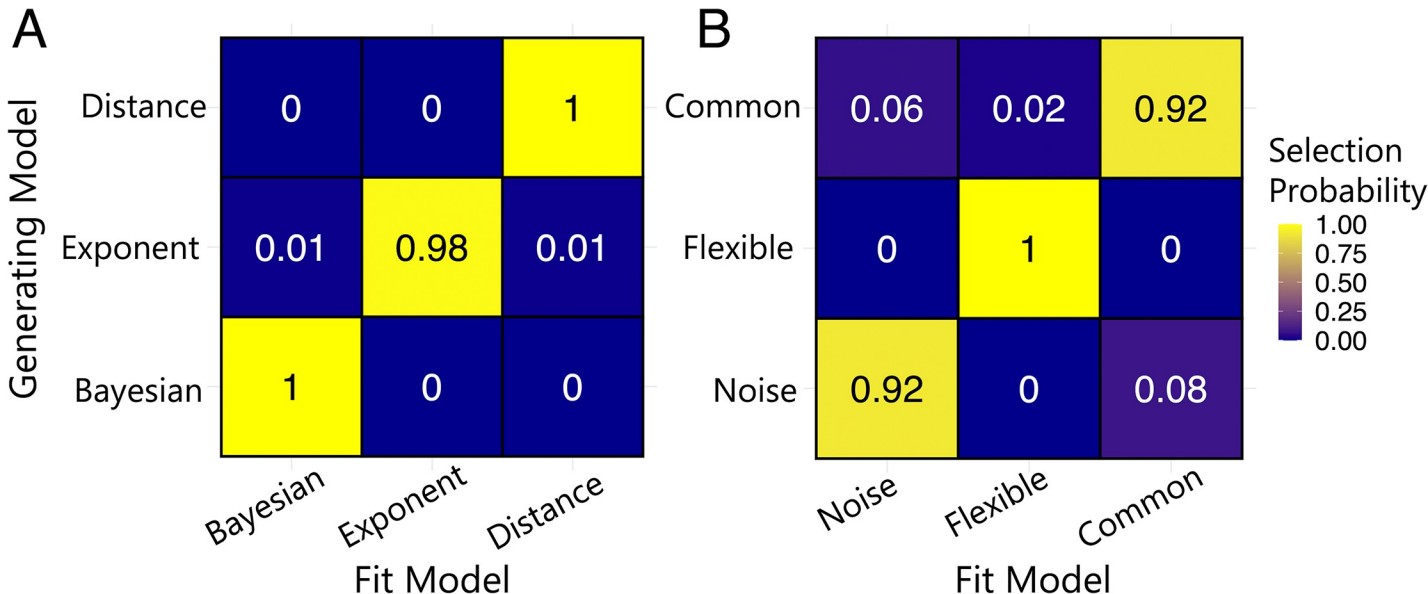

**Fig 8. Model Recovery.** Numbers and colours denote the probability that the data generated with model X (x-axis) are best fit by model Y (y-axis). Selection probabilities were calculated using *AIC* for model selection. (A) Confusability matrix for core models using a representative model from each model class. (B) Confusability matrix for models used to compare parameter settings across modalities.

## Discussion

This study investigated whether the computations involved in metacognitive judgements generalise across sensory modalities. Participants completed auditory and visual versions of a categorisation task where evidence strength and sensory uncertainty were varied in each modality. We used computational modelling to quantify the mapping of evidence strength and sensory uncertainty to confidence, evaluating the performance of three distinct classes of confidence models: unscaled evidence strength, scaled evidence strength and Bayesian. We found that a single class of models, namely the scaled evidence strength models, provided the best account of category and confidence judgements in both the visual and auditory domain but that the parameters governing this process are tuned to different values across modalities. Here we discuss how our findings provide insight into the computations underlying metacognitive judgements across modalities and the extent to which these computations operate in a modality-independent or modality-specific manner.

### The computations underlying metacognitive judgements across modalities

**The importance of accounting for sensory uncertainty.**   Our model comparison results indicated that observers modulate their confidence with sensory uncertainty, not just evidence strength, in both the visual and auditory domain. Specifically, the unscaled evidence strength models, in which confidence depends entirely on the strength of the signal, performed consistently worse than the other model classes across all task-modality configurations. Unlike the other model classes, the unscaled evidence strength class did not independently quantify sensory uncertainty, such that a given distance between the decision variable and the decision criterion was always associated with the same level of confidence [26–28]. This finding is particularly relevant because the importance of sensory uncertainty for confidence has not yet been demonstrated in the auditory modality. Thus, we were able to provide evidence that observers take sensory uncertainty into account across modalities, contributing to our limited

understanding of the computational basis of confidence across domains. Furthermore, our study provides insight into the extent to which these sensory-uncertainty-dependent processes are consistent with Bayesian or non-Bayesian accounts of confidence.

**Implications for Bayesian models of confidence.**   Although we found consistent evidence that observers take sensory uncertainty into account, we observed significant differences between our empirical data and the predictions of the Bayesian models. In our tasks, the Bayesian class of models assumed that observers use knowledge of the level of sensory uncertainty associated with the measurement of the stimulus *and* (sometimes veridical) knowledge of the generating category distributions to compute a log posterior probability ratio for category membership. The log posterior probability ratio was then compared with a set of thresholds in log posterior probability ratio space to determine confidence. We found that this class of models was outperformed by the scaled evidence strength class, both for the group as a whole and for all individual participants. The relatively poor performance of the Bayesian models did not seem to be due to lack of flexibility. The Bayesian models continued to perform poorly even after allowing for a noise term in the calculation of the log posterior probability ratio or inaccurate subjective beliefs about the generative model (i.e., freely estimated category distribution parameters).

While the poor performance of the Bayesian models is in disagreement with some studies [3, 15, 37, 40, 42, 43], it is consistent with others [3, 29, 33, 44, 45, 48]. Importantly, however, our study extends existing research on Bayesian models of confidence in two ways. First, most previous studies have focused exclusively on the visual domain. Here we compared Bayesian and non-Bayesian accounts of confidence in both the visual and auditory domains, thus showing converging evidence for non-Bayesian metacognitive computations in a broader range of sensory contexts. Second, most previous studies have used more simple task structures. Here we found that the Bayesian class did not perform as poorly when there was a single clearly defined category boundary, as in the different means task. In the different SDs task, however, where the categories overlapped to a greater extent and optimal categorisation required observers to consider the variability of both categories, the Bayesian models performed much worse. This suggests that the support for Bayesian computations in previous research may be partly attributed to low task complexity. Because the computations involved in confidence generation become increasingly complex as the task structure becomes more complex (i.e., tasks where there is a high overlap of exemplars from different categories), deviations between human behaviour and predictions from Bayesian principles may become more pronounced. Our findings suggest that future research evaluating Bayesian models should rely more on complex category structures, that likely reflect the dynamic conditions encountered in the real world, to assess human behaviour relative to Bayesian optimality.

Although our study used a highly representative set of Bayesian models from the broader confidence literature, we could not feasibly test the full space of possible models that might be considered "Bayesian". We, therefore, cannot rule out the entire Bayesian class of confidence models completely. In particular, our Bayesian models may have been limited by only incorporating a point estimate of sensory uncertainty, rather than a distribution over uncertainty [34]. Furthermore, we did not consider models with non-Gaussian noise, lapse rate parameters or non-Gaussian priors. Our findings, nonetheless, strongly suggest confidence judgements in the visual and auditory modalities are unlikely to derive from posterior probability computations as defined by the set of Bayesian models considered here. Thus, although Bayesian models are particularly powerful for their generalisability, that is they involve the computation of a probability metric which can be compared directly across domains, their poor fit to empirical data highlights the need to explore non-Bayesian frameworks that can better capture human behaviour in different contexts. Our study, in particular, demonstrates that non-Bayesian

algorithms such as the scaled evidence strength models can operate on standardised units and provide a powerful account of human confidence judgements across modalities.

**Non-Bayesian sensory uncertainty dependent computations.**   We found that the scaled evidence strength class of models, in which positioning of the confidence and choice criteria depended on a point estimate of sensory uncertainty but did not involve the computation of posterior category probabilities, provided the best account of category and confidence responses across modalities [29, 30, 33]. We found consistent evidence for this class of models both at the group level, where the models were best able to capture the main patterns of behaviour across participants in both modalities and tasks, and at the individual participant level, where the same class of models was preferred for all participants across modalities and tasks. Using response criteria that depend on an estimate of sensory uncertainty allows observers to avoid assigning high confidence to highly variable stimuli [33, 49], such that choice/confidence boundaries can be compressed (or expanded, where appropriate) under conditions of high sensory noise and vice versa under conditions of low sensory noise. This sort of algorithm produces behaviour that is similar to the optimal Bayesian observer but is less computationally expensive, as it does not involve the computation of the posterior probability distribution [29].

**A Scaling Factor for Sensory Uncertainty.**   Although we found that observers use sensory uncertainty-dependent boundaries for category and confidence judgements, the specific way sensory uncertainty scales the boundaries was not universal across modalities or tasks. Within the scaled evidence strength class, we tested different assumptions about the scaling rule that was used to adjust the confidence criteria. We compared models which used either a linear ($k + m\sigma$), quadratic ($k + m\sigma^2$), or free-exponent ($k + m\sigma^a$) scaling rule. We estimated a value of $k$ and $m$ for each boundary, which allowed for the strength of the effect of sensory uncertainty on the positioning of the boundaries to differ across response options (i.e., in the different means tasks, the confidence 4 boundary could shift more than the confidence 1 boundary). In the case of the linear scaling rule, $m$, was a constant that was applied to the estimate of sensory uncertainty which determined the magnitude of the sensory-uncertainty-based shift in choice/confidence boundaries. In the cases of the non-linear scaling rules, an exponent was also used on the estimate of sensory uncertainty assuming that the impact of sensory uncertainty on confidence criteria was not proportional to the amount of sensory uncertainty. Rather, these models assumed that the impact of sensory uncertainty became more pronounced at higher levels. These different models allowed us to compare the relative importance of different scaling rules which quantified different assumptions about the effect of sensory uncertainty on choice/confidence criteria. We found, however, that both the linear and non-linear scaling rules accounted for the empirical data well and we did not find consistent evidence in favour of either alternative.

The lack of consistency within the scaled evidence strength class suggests that the scaling of sensory uncertainty itself is important, but that the scaling factor depends on contextual factors such as modalities, task structure and individuals. Thus, it is likely that the scaling rule used to update confidence criteria reflects properties such as task difficulty or stimulus familiarity, rather than reflecting qualitative differences in the computational processes used by individuals within or across modalities. From a measurement perspective, this supports the case for freely estimating the scale factor, as it potentially provides a better assessment of how people's confidence judgments are affected by sensory uncertainty. Future research might benefit from using this approach to investigate what contextual factors moderate different strategies (and scale factors) for adjusting response criteria by an estimate of sensory uncertainty.

## Future directions and next steps for computational models of confidence

Although our tasks were carefully designed to allow for comparison across modalities, we believe that the scaled evidence strength class of models can generalise beyond the narrow conditions investigated here. In our tasks, the decision variable for the scaled evidence strength models was operationalised as a noisy measurement of the stimulus feature of interest (standardised orientation or frequency). For other tasks, however, the decision variable may represent a more complex, multi-dimensional signal. For example, the decision variable could represent sensory information that has been integrated across domains (i.e., an audio-visual signal), utility in a value-based decision-making task or any other normalised variable in a range of decision-making tasks. Our results strongly suggest that this class of models provides a promising direction for future research to investigate a generalisable algorithm for confidence.

Of note, we did not consider models which jointly account for choice, confidence, and response time. Given the established link between response time and confidence [32, 35, 40, 50–54], we see a formal investigation of the generalisability of response time-based models across domains as an important field of investigation for future research. Our study serves an important role in narrowing the scope of potential models and provides several important considerations for future studies investigating response time. In particular, our results show that in addition to the strength of the signal, the amount of sensory uncertainty in the signal plays an important role in the formation of confidence, beyond just adding noise to the decision-making process. This is true across different decisional domains, that is, different task structures and modalities.

## Implications for modality-independent or modality-specific metacognition

Our results suggest that the general algorithm used for relating choice and confidence is the same across modalities. We did not find, however, that this process was completely 'modality-independent'. We sought to determine the extent to which the scaled evidence strength algorithm operated in a "one size fits all" configuration or whether it required different parameter settings to accommodate responding across modalities. We fit the free-exponent model to data from both modalities simultaneously and formulated several levels of flexibility in the parameter settings of the model. The most flexible version of the model allowed the parameter values to vary across modalities, and the less flexible versions constrained parameters to be the same across modalities. We found that the most flexible model, in which all parameters were estimated independently for each modality, was the best performing model both at the individual level and the group level. Therefore, although the scaled evidence strength algorithm provided the best account of category and confidence judgements across both tasks and modalities, it is likely this process is tuned/parameterised specifically in each sensory domain. The existence of a general confidence process which may be tuned specifically to task requirements is consistent with studies finding both 'domain general' and 'domain specific' components of metacognition [7, 46, 55–58]. These studies suggest that generic confidence signals are combined with domain-specific information to fine-tune decision making [57, 59].

The existence of a general set of computations for confidence which are fine-tuned within a domain may be particularly relevant when making comparative judgements across modalities or making decisions about multidimensional stimuli. For example, where an observer has to compare their confidence in a visual decision to an auditory decision or an observer has to integrate uncertainty from both the visual and auditory dimensions of a stimulus to assess their confidence. Having a common currency for confidence facilitates the integration and comparison of information across dimensions, which might maximise processing efficiency

[24, 25]. Importantly, however, we did not require participants to combine sensory information across modalities in our tasks. Future research should investigate the underlying metacognitive computations when participants are required to do so to validate the generalisability of the findings presented here.

## Conclusions

In conclusion, our study provides strong evidence that of the existing set of popular confidence models, the scaled evidence strength class provides the best account of confidence in both our visual and auditory categorisation tasks. Our model comparison results provide insight into the metacognitive computations used across modalities and the extent to which these processes operate in a modality-independent configuration. Our study paves the way for future research investigating how these computations generalise to other decisional domains (e.g., multi-dimensional sensory signals or value-based decision-making), vary with contextual factors, and how these algorithms may be adapted to account for response time.

## Methods

### Ethics statement

This study was approved by the University of Queensland Health and Behavioural Sciences, Low & Negligible Risk Ethics Sub-Committee (#2020001811). All participants provided written consent prior to participation.

### Participants

12 participants ($M_{age}$ = 24.83, $SD_{age}$ = 4.35) were recruited through The University of Queensland's research participation scheme and by word-of-mouth. Participants were reimbursed for their time ($20 per hour in cash or gift cards). Inclusion criteria included normal (or corrected-to-normal) vision and normal hearing, both assessed by self-report.

### Overview

Participants completed two categorisation tasks in each of two sensory modalities, vision and audition (all factorial combinations are collectively referred to as *task-modality configurations*). On each trial, participants made a two-alternative forced choice category decision, followed by a 4-point confidence rating, ranging from low confidence (1) to high confidence (4). The categorisation tasks were based on those used by Adler and Ma (2018). Category membership was determined by a single stimulus attribute: orientation for visual stimuli (drifting Gabor patches) and pitch (frequency) for auditory stimuli (pure tones). A normal distribution of the relevant stimulus attribute defined each category and the parameters of these category distributions differed across two versions of the task: a *different means* task and a *different standard deviations* (SDs) task. In the different means task, category distributions had different means and the same standard deviation. In the different SDs task, category distributions had the same mean and different standard deviations. The category distributions are described in detail below. In both tasks, sensory uncertainty was also varied for each modality, with stimuli presented at 4 different intensities. In the visual tasks, intensity depended on the contrast of the Gabor patch (3.3%, 5%, 6.7%, 13.5%). In the auditory tasks, intensity depended on the loudness of the tone (4, 9, 26, 55 phon; phons are the sound pressure level in decibels of a 1000 Hz pure tone that has been subjectively matched in loudness to a target tone).

## Categorisation tasks

**Different means task.** As shown in **Fig 2**, category distributions in the different means task had different means and the same standard deviation. In the visual modality, where 0 degrees represented a horizontal Gabor patch, orientations that were rotated counter clockwise relative to horizontal (i.e., negative orientations) were more likely to be sampled from category 1 ($\mu_{cat1} = -4˚$, $\sigma_{cat1} = 5˚$). Orientations that were rotated clockwise relative to horizontal (i.e., positive orientations) were more likely to be sampled from category 2 ($\mu_{cat2} = 4˚$, $\sigma_{cat2} = 5˚$). In the auditory modality, the category distributions had the same properties, but parameter values were shifted into the frequency domain. Lower frequency tones were more likely to be sampled from category 1 ($\mu_{cat1} = 2300\ Hz$, $\sigma_{cat1} = 475\ Hz$) and higher frequency tones were more likely to be sampled from category 2 ($\mu_{cat2} = 3100\ Hz$, $\sigma_{cat2} = 475\ Hz$).

Stimuli with orientation/frequency values equal to where the two category distributions intersected, at 0 degrees for orientations and 2700 Hz for frequencies, were equally likely to be sampled from either category 1 or category 2. Because the relative likelihood of category membership changed monotonically around this point, to maximise correct responses, an ideal observer would report that any stimulus value below this point was sampled from category 1 and any stimulus value above this point was sampled from category 2.

**Different SDs task.** In the different SDs task, category distributions had the same mean and different standard deviations. In the visual modality, larger rotations clockwise or counter clockwise relative to horizontal were more likely to be sampled from category 2 ($\mu_{cat2} = 0˚$, $\sigma_{cat2} = 12˚$) whereas smaller rotations relative to horizontal were more likely to be sampled from category 1 ($\mu_{cat1} = 0˚$, $\sigma_{cat1} = 3˚$). In the auditory modality, lower and higher frequency tones were more likely to be sampled from category 2 ($\mu_{cat2} = 2700\ Hz$, $\sigma_{cat2} = 500\ Hz$), whereas intermediate frequency tones were more likely to be sampled from category 1 ($\mu_{cat1} = 2700\ Hz$, $\sigma_{cat1} = 125\ Hz$).

To maximise correct responses, an ideal observer would use the two points where the category distributions intersect, at -5 and 5 degrees for orientations and 2485 Hz and 2915 Hz for frequencies, reporting that stimulus values contained within these intervals were sampled from category 1 and stimulus values outside these intervals were sampled from category 2.

## Stimuli

**Visual stimuli.** The visual stimuli were drifting Gabors which had a spatial frequency of 0.5 cycles per degrees of visual angle (dva), a speed of 6 cycles per second, a Gaussian envelope with a standard deviation of 1.2 dva, and a randomized starting phase. Visual stimuli appeared at fixation for 50 ms.

**Auditory stimuli.** The auditory stimuli were tones of varying frequency (including 5 ms linear onset and offset amplitude ramps to eliminate onset and offset clicks). Stimuli were synthesized with an ASIO4ALL sound driver with a sampling rate of 28 kHz. Tones were matched for loudness according to the International Standard ISO 226:2003: Acoustics- Normal Equal-Loudness-Level-Contours (International Standardization Organisation, 2003) and extensive piloting was undertaken to select intensity levels. Auditory stimuli were presented in stereo via headphones (Sennheiser HD 202, see **S7 Text** for calibration details) for 50 ms.

## Procedure

All participants completed all combinations of tasks in four separate testing sessions (i.e., visual different means task, visual different SDs task, auditory different means task, auditory different SDs task). Participants completed the *different means* and the *different SDs* task of the same modality on the same day (with at least one hour between sessions) and completed the

visual and auditory tasks on separate days. The order of task modality was counterbalanced across participants and task type (different means vs. different SDs) was counterbalanced within modalities and across participants.

Participants were seated in a dark room, at a viewing distance of 57 cm from the screen, and their head was stabilised with a chinrest. Stimuli were presented on a gamma-corrected 60 Hz 1920-by-1080 display. The computer display (ASUS VG248QE Monitor) was connected to a Dell Precision T1700, calibrated with a ColorCal MKII (Cambridge Research Systems). Stimuli were generated and presented using custom code and the Psychophysics Toolbox extensions [60, 61] for MATLAB.

In each session, participants received instructions for that session's task, which included an explanation about how the stimuli were generated from the relevant category distributions. To further illustrate these distributions, participants were then shown 36 stimuli randomly sampled from each category. They then completed the category training and testing (detailed below). Each session took approximately 1 hour. Combining all sessions and tasks, participants completed 1440 training trials and 2880 testing trials. Data from training trials were not included in any analyses.

**Category training.**   At the start of each training trial, participants fixated on a central cross for 1 s. The fixation cross was then extinguished, and a stimulus was presented. A stimulus value (i.e., orientation for visual stimuli and frequency for auditory stimuli) was sampled randomly from the relevant category distribution and the stimulus was presented (50 ms duration for auditory stimuli and 300 ms duration for visual stimuli). Immediately after the offset of the stimulus, participants were able to respond category 1 or category 2 by pressing the F or J key on a standard keyboard with their left or right index finger, respectively. No confidence ratings were collected during training. After participants made their response, corrective feedback (i.e., the word "correct" in green or "incorrect" in red) was displayed for 1.1 s. The intertrial interval was 1 s, after which the fixation cross reappeared. Within a training block, stimuli were sampled equally from each category and the order of categories was randomised across trials.

Participants completed 3 training blocks (120 trials per block, in total 360 trials) per session. During training, only the highest intensity level was used for stimulus presentation.

**Test.**   The trial procedure in testing blocks was the same as in training blocks, except that trial-to-trial feedback was withheld, stimuli were presented at four different intensity levels (different contrast values for the visual stimuli and different loudness values for the auditory stimuli) and stimuli were presented for 50 ms, regardless of modality. Participants then made a category response, followed by a confidence report, using the 1–4 number keys to indicate their confidence. We chose to use a 4-point confidence rating scale as it is common in the study of confidence and allowed for meaningful comparison of models to previous research [29, 48] (see [62] for investigation of alternative methods for collecting confidence ratings).

At the end of each block, participants were required to take at least a 30 s break. During the break, they were shown the percentage of trials they had correctly categorized in the most recent block. Participants were also shown a list of the top 10 block scores (across all participants, indicated by initials) for the task they had just completed. This was intended to motivate participants to maintain a high level of performance. Participants completed 6 testing blocks per session (120 trials per block, a total of 720 trials per session). Within a testing block, an equal number of test stimuli were sampled from each category and each intensity level (i.e., 15 trials per cell of the design). This meant that within a testing block, participants saw stimuli from both categories presented at every intensity level. The order of both category and intensity level were randomised.

## Model specification

To model responses, we assumed that on each trial, *i*, the observer receives a sensory signal, *x*. The observer transforms that sensory signal into a continuous decision variable and then compares this variable to a set of *boundaries* to make a discrete category-confidence response. In the following sections, we describe our stimulus standardisation procedure, our assumptions about the sensory signals, decision variable, decision boundaries and then describe the model fitting procedure.

## Standardisation of stimulus values

To directly compare stimulus values and estimated model parameters across modalities, we standardised stimulus values before model fitting. For each stimulus value in each task-modality configuration, we subtracted the mean and divided by the standard deviation of all stimulus values such that:

$$Z = \frac{s - \mu_{[M,T]}}{sd_{[M,T]}} \tag{3}$$

where *M* represents a given modality (visual or auditory); *T* represents a given task (different means or different SDs); *Z* is the standardised stimulus value; *s* is the true stimulus value in perceptual units (degrees or hertz); $\mu$ is the mean of all presented stimulus values for that task and modality; and *sd* is the standard deviation of the same set of stimulus values. The standardisation of sensory units is consistent with the idea that participants built up a reliable internal representation of the categories during training such that they understood the relative frequencies of different stimulus values within each category distribution. See **Fig 9** for visualisation of standardised stimulus values.

## Sensory signals

**Measurement Noise.**   In all models, we assume that the observer's internal representation of the stimulus, *x*, is a noisy measurement. We approximate this measurement noise using a Gaussian distribution (referred to as the 'measurement distribution') in both modalities. Although orientation is circular and traditionally modelled using a von Mises distribution, we used a Gaussian distribution because we only used a small range of orientations in both the visual tasks [29]. The Gaussian distribution was centred on the true stimulus value presented on trial *i*, $s_i$, with standard deviation, $\sigma$. We assume that $\sigma$ depended on stimulus intensity, where greater stimulus intensity was associated with less measurement noise. In contrast to Adler and Ma (2018), we did not enforce a power law relationship between intensity and measurement noise because we were agnostic about the functional form of this relationship. Therefore, the standard deviation of the measurement distribution was estimated separately at each intensity level with a monotonicity constraint,

$$x \sim N(s_i, \sigma_i^2) \tag{4}$$

where *N* denotes the normal density function. The standard deviation of the measurement distribution, therefore, approximated the level of sensory uncertainty in the observer's perception of the stimulus.

**Orientation Dependent Noise.**   Following Adler and Ma (2018), for modelling performance in the visual modality, we also tested a variation of some models which assume additive orientation-dependent noise in the form of a rectified 2-cycle sinusoid [63]. In these models,

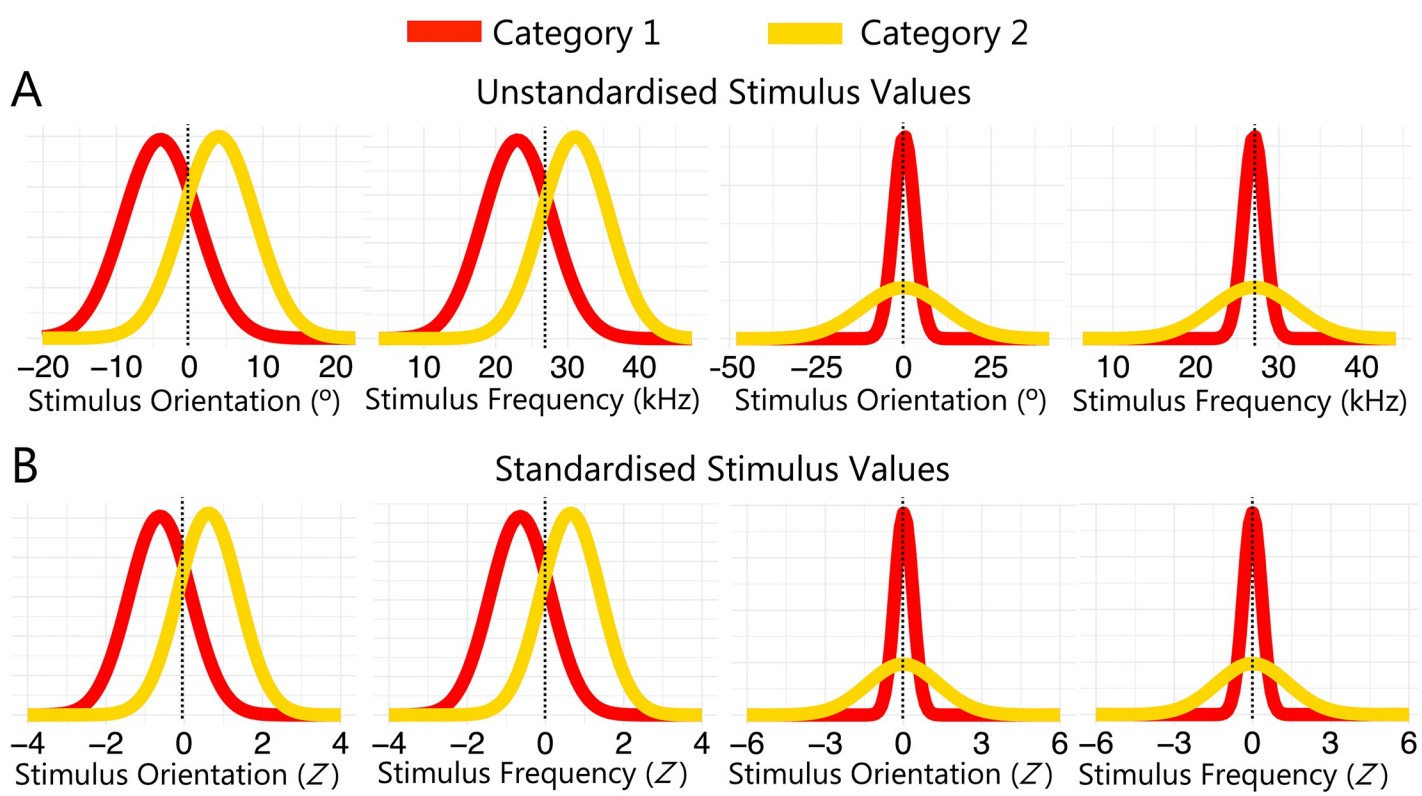

**Fig 9. Presented Stimulus Values Before and After Standardisation for All Tasks.** (A) Presented stimulus orientations (visual modality) and stimulus frequencies (auditory modality) across different means (left panels) and different SDs (right panels) tasks before standardisation. Note similarity in presented stimulus distribution with true generating distributions in **Fig 2**. (B) Presented stimulus values after standardisation. Stimulus values were standardised by subtracting the mean and dividing by the standard deviation for all presented stimulus values, separately for each of the task-modality configurations.

the standard deviation of the measurement distribution was:

$$\sigma_i(I, s) = \gamma_I + \psi |\sin\frac{\pi s}{90}|, \tag{5}$$

where $I$ represents the intensity level, $\gamma_I$ is a free parameter representing the baseline amount of measurement noise at a given intensity and $\psi$ scales the amount of additive measurement noise across stimulus orientations. We did not fit a frequency dependent noise parameter in the auditory tasks as each frequency was corrected for equal loudness prior to adjusting overall intensity.

### Decision boundaries across tasks

In all models, we assumed that the observer transformed the sensory signal into a decision variable which was then compared to a set of *boundaries* to make a discrete category-confidence response. The boundaries partition the stimulus space into discrete category and confidence response regions so that if a stimulus falls within the defined region, response $r$ is given. Given four confidence levels for each of the two categories, we required eight response regions, $r \in \{1,2,3,\ldots,8\}$, each corresponding to a unique category and confidence combination. Given the generating category distributions (see **Fig 2**), the positioning of these boundaries differs for the different means and the different SDs tasks and is described below.

**Different means task boundaries.** In the different means task, we assumed that one of the boundaries, $b_1$, divides the stimulus range into the two different category response regions. That is, where the observed stimulus value is below this boundary, a category 1 response is given and where the observed stimulus value is above this boundary, a category 2 response is given. Confidence associated with the category response is determined by a further three boundaries, $b_2$, $b_3$, $b_4$, which created four different confidence response regions within each category. Within a category response region, we assumed that the confidence boundaries increased monotonically across successive boundaries such that higher confidence response regions were aligned with more extreme category evidence (see **Fig 4B**). We also assumed that the confidence boundaries were symmetrically positioned across categories. Due to symmetry, confidence regions on equidistant but opposite sides of the category boundary, $b_1$, were treated as equivalent by the model. The boundaries in the different means task were ordered such that: $-b_4 < -b_3 < -b_2 < b_1 < b_2 < b_3 < b_4$, where the position of each boundary was freely estimated, but constrained by monotonicity.

**Different SDs task boundaries.** In the different SDs task, two boundaries, $-b_4$ and $b_4$, divided the stimulus range into two different category response regions. Where a stimulus value falls within the $-b_4$ and $b_4$ region, a category 1 response is given. Where a stimulus value falls beyond $-b_4$ or $b_4$, a category 2 response is given. To make a confidence response, the observer uses 4 symmetrical boundaries for each category: $b_1-b_4$ for category 1 responses and $b_4-b_7$ for category 2 responses, as shown in **Fig 4B**. In all models, we assumed that the boundaries increased monotonically across successive boundaries such that higher confidence response regions were aligned with more extreme category evidence (see **Fig 4B**). We also assumed that the boundaries are symmetrically positioned around the means of the category distributions. For ease of exposition, we refer to this centre point as $b_0$. We assumed the boundaries are symmetrical in this task because two measurements that are equidistant from the middle of the category structure but on opposite sides of $b_0$ should logically have the same confidence associated with them. Due to symmetry, confidence regions on equidistant but opposite sides of $b_0$ were treated as equivalent by the model. The boundaries in the different SDs task were ordered such that:

$$-b_7 < -b_6 < -b_5 < -b_4 < -b_3 < -b_2 < -b_1 < b_0 < b_1 < b_2 < b_3 < b_4 < b_5 < b_6 < b_7.$$

## Decision variables across models

We tested different assumptions about the transformation of the sensory signal into the decision variable. These assumptions map onto different theoretical models about the choice and confidence generation process. We categorised these models into three broad classes: the unscaled evidence strength models, the scaled evidence strength models and the Bayesian models. One of the distinguishing factors among candidate models concerns how the decision variable and the category-confidence boundaries were positioned in either perceptual space (scaled and unscaled evidence strength models; discussed in more detail below) or probability space (Bayesian models; discussed in more detail below) as a function of sensory uncertainty.

For models that represent the decision variable in perceptual space, the resulting boundaries were in standardised units that related directly to the stimulus and, therefore, the sensory signal could be compared directly with the boundaries. For these models, we use $x$ to denote both the sensory signal and the decision variable. For models that represent the decision variable in probability space, the observer's internal representation of the stimulus was transformed into a probability metric which was then compared with the boundaries. For these models, we use $d$ to denote the decision variable and distinguish it from the sensory signal. To illustrate, for the different means task in the visual domain, for a non-Bayesian model with

category-confidence boundaries in perceptual space, a Category 2 Confidence 1 response might be made for any stimulus perceived as having an orientation between 0 and 5 degrees (or 0 and 0.6 in standardised stimulus units). For a Bayesian model, where the boundaries are in probability space, the same Category 2 Confidence 1 response might be made for any stimulus perceived as having a log posterior probability ratio between 0 and 0.5.

A description of the distinguishing elements of each model class is provided below. All models were fit to each task and modality separately (except where specified otherwise).

### Non-Bayesian models

The non-Bayesian models constituted models from both the scaled evidence strength and unscaled evidence strength class. For all these models, we refer to the position of the boundaries in perceptual space as $b$. For the unscaled evidence strength models and the scaled evidence strength models, we made different assumptions about how $b$ varies as a function of sensory uncertainty ($\sigma$).

**Unscaled evidence strength models: Distance Model.**   In the distance model, the boundary positions were not dependent on (estimated) values of sensory uncertainty ($\sigma$) and therefore occupied fixed positions across intensity levels. The position of each boundary in perceptual space was described by a free parameter, $k$, where perceptual boundaries $b_r = k_r$.

**Scaled evidence strength models: Linear, Quadratic and Free-Exponent Models.**   In the linear and quadratic models, the positions of the boundaries were linear or quadratic functions of $\sigma$. Each boundary position depended on 3 free parameters: $k$, $m$ and $\sigma$. $k$, represented the base position of each boundary which was offset by a scaled value of sigma, $\pm m\sigma$ (linear model), or by a scaled value of sigma squared, $\pm m\sigma^2$ (quadratic model), where $m$ was fixed for each boundary. The resulting boundaries in perceptual space were given by $b_r(\sigma) = k_r + m_r\sigma$ (Linear) and $b_r(\sigma) = k_r + m_r\sigma^2$ (Quadratic). In the free-exponent model, we estimated the exponent on σ, $a$, as a free parameter, $b_r(\sigma) = k_r + m_r\sigma^a$. The additional free parameter allowed for the changes in boundary position across intensities to occupy values between and beyond linear or quadratic values. We constrained $a$ to be greater than 1.

**Parameter settings of the free-exponent models across modalities.**   While the primary focus of this study was to determine whether a single class of metacognitive models could account for the way participants make confidence judgements across modalities, we also wanted to determine the extent to which this process was sensitive to different parameter settings. Accordingly, rather than fit the model to each modality separately, we fit the free-exponent model to data from both modalities simultaneously. We formulated several levels of flexibility in the parameter settings, with more flexible versions allowing the parameter values to be different across modalities and less flexible versions constraining parameter values to be the same across modalities. We chose the free-exponent model to test the flexibility in parameter settings, as it was the most flexible of the scaled evidence strength models and included the linear and quadratic models as special cases.

*Common settings model.*   For the common settings model, both modalities had the same parameter settings. We assumed that sensory uncertainty ($\sigma$) did not differ across modalities. Thus, we fit $\sigma$ as a free parameter for each intensity level and used the same value of $\sigma$ across modalities. We also assumed that the same boundary positions were used across modalities so that all perceptual boundaries were given by $b_{r[I]}(\sigma) = k_r + m_r\sigma_{[I]}{}^a$ where $I$ is the intensity level.

*Different noise settings model.*   For the different noise settings model, sigma was a free parameter for each intensity level in each modality. $k$ and $m$ were fixed across modalities, assuming that only measurement noise changed across modalities. The perceptual boundaries were given by $b_{r[I,M]} = k_r + m_r\sigma_{[I,M]}^a$, where $I$ is the intensity level and $M$ is the sensory modality.

***Flexible settings model.*** In the flexible settings model, all parameters in the standard free-exponent model were freely estimated in each modality.

## Bayesian models

Consistent with Bayesian decision theory, we assumed that a Bayesian observer optimally combines prior beliefs of the generative model (i.e., the task-specific category distributions) with observed stimulus information to form posterior beliefs about the nature of the stimulus (i.e., category membership). For all Bayesian models, therefore, we assume that the observer transformed the sensory signal into a probability metric, *d*. We tested several versions of Bayesian confidence models which make different assumptions, as described below.

**Log posterior probability ratio.** When the perceived value of a stimulus on a given trial is *x*, we assume that the observer uses the log posterior probability ratio to make a decision. The log posterior probability ratio represents the (log) ratio of posterior beliefs about category membership, given the properties of the stimulus, such that $d = log\frac{p(C=1|x)}{p(C=2|x)}$. When the log posterior probability ratio is positive, there is greater evidence for category 1. We assumed that category and confidence reports depended on the sensory signal, *x*, only via *d*. On a given trial, the observer chooses a response by comparing *d* to a set of boundaries $k = (k_1, k_2 \ldots k_7)$ that partitions *d* space into eight response regions, each representing a unique category and confidence combination. The positions of these boundaries in *d* units were free parameters. Following Adler and Ma (2018; see **S8 Text**), we used the following task-specific derivations of *d* for the different means task:

$$d_{different\ means} = \frac{2x\mu_1}{\sigma^2 + \sigma_1^2} + log\frac{p(C=1)}{p(C=2)} \tag{6}$$

and the different SDs task:

$$d_{different\ SDs} = \frac{1}{2}log\frac{\sigma^2 + \sigma_2^2}{\sigma^2 + \sigma_1^2} - \frac{\sigma_2^2 - \sigma_1^2}{2(\sigma^2 + \sigma_1^2)(\sigma^2 + \sigma_2^2)}x^2 + log\frac{p(C=1)}{p(C=2)} \tag{7}$$

where $\mu_1$ and $\sigma_1$ are the mean and standard deviation of the category 1 distribution, $\sigma_2$ is the standard deviation of the category 2 distribution and $\sigma$ is the fitted value of measurement noise, approximating the amount of sensory uncertainty in the observers' internal representation of the stimulus (see Eq 4). In terms of the generative model, in the different means task the observer needs knowledge of $\mu_1$ and $\sigma_1$ to make an optimal decision and in the different SDs task the observer needs knowledge of $\sigma_1$ and $\sigma_2$ to make an optimal decision. Because we sampled equally from each category, we assumed that $p(C = 1) = p(C = 2) = 0.5$ such that $log\frac{p(C=1)}{p(C=2)}$ is 0 in both Eq 6 and Eq 7.

**Log posterior probability ratio with decision noise.** For the log posterior probability ratio model with decision noise, we assumed that there was an added Gaussian noise term on *d*, $\sigma_d$. This allowed for the observer's calculation of *d* to be noisy, in addition to sensory noise in the internal representation of the stimulus, $\sigma_i$ [29].

**Log posterior probability ratio with free category distribution parameters.** The log posterior probability ratio model assumes that the observer knows the true values of the means and standard deviations of the category distributions, $\mu_1$ and $\sigma_1$ in Eq 6 and $\sigma_1$ and $\sigma_2$ in Eq 7. We also tested a version of the model in which the observer had imperfect knowledge of the parameters of the generative model, and estimated these values as free parameters instead.

**Fitting the models to data.** Model parameters were estimated for each participant using maximum likelihood estimation (see **S9 Text**). Below we describe the process for calculating

the likelihood of the dataset, given a model with parameters $\theta$, for the non-Bayesian and Bayesian models, as they require different transformations.

**Non-Bayesian models.** For all non-Bayesian models, the boundaries were positioned in perceptual space and no transformation of boundary positions was required. For each trial, we determined the predicted response probability for each combination of category and confidence level by computing the probability mass of the measurement distribution between adjacent boundaries, given the parameters of the model and the participant's response on that trial, $r_i$ (cf. [64]; see **Fig 4D**). For the different means task, this quantity is:

$$\int_{b_{r_i-1}}^{b_{r_i}} N(x;\ s_i, \sigma_i^2) dx, \tag{8}$$

where $b_5 = \infty$ and $-b_5 = -\infty$. For the different SDs task, this quantity is:

$$\int_{-b_{r_i}}^{-b_{r_i-1}} N(x;\ s_i, \sigma_i^2) dx + \int_{b_{r_i-1}}^{b_{r_i}} N(x;\ s_i, \sigma_i^2) dx \tag{9}$$

where $b_0 = 0$, $b_8 = \infty$ and $-b_8 = -\infty$.

**Bayesian models.** In the Bayesian models, boundaries are positioned in $d$-space (log posterior probability ratio space). To evaluate the probability mass of the measurement distribution, we transformed the boundary positions into perceptual space, $b(\sigma)$, using parameters $k$ as the left-hand side of Eq 6 or Eq 7 and solved for $x$ at the fitted levels of $\sigma$. Solving Eq 7 for $x$ requires calculating the square root of $d$, which is undefined for negative values. We therefore used the absolute value of $d$ to solve Eq 7 for $x$ and applied the appropriate sign to the result to arrive at a final standardised orientation/frequency value for $x$. Where the category distribution means and standard deviations were free parameters, we used the estimated values in place of the true mean and/or standard deviation/s ($\mu_1$ and $\sigma_1$ for the different means task and $\sigma_1$ and $\sigma_2$ for the different SDs task). We then calculated the probability mass of the measurement distribution between adjacent perceptual boundaries using Eq 8 or Eq 9, given the task and participant's response for that trial.

**Log Posterior Probability Ratio with Decision Noise.** For the log posterior probability ratio model with decision noise, we took 101 evenly spaced draws from a normal distribution (spanning the 1st to 99th distribution percentiles) for each boundary, with $k$ as the mean and the standard deviation as a free parameter, $\sigma_d$. Each of the 101 draws was then converted into perceptual space using Eq 6 or Eq 7 to solve for $x$ at the fitted levels of $\sigma$. We then calculated the probability mass of the measurement distribution between corresponding draws of the relevant boundaries, given the participant's response for that trial. This meant that for a single trial we had 101 response probabilities. We calculated the weighted sum of these probabilities using the normalised densities for each draw of $k$.

**Log likelihood of the dataset.** To obtain the log likelihood of the dataset, we computed the sum of the log probability for every trial $i$, where $t$ is the total number of trials.

$$\log p(data|\theta) = \sum_{i=1}^{t} \log p(r_i|\theta) = \sum_{i=1}^{t} \log p_\theta(r_i|s_i, \sigma_i) \tag{10}$$

## Model selection

To compare candidate models, we used Akaike information criterion (*AIC*) and Bayesian information criterion (*BIC*) for model selection. For a single participant, *AIC* is defined as:

$$AIC = 2z - 2\log(L) \tag{11}$$

where $z$ is the number of model parameters and $\log(L)$ is the log likelihood of the dataset. To quantify support for each model across all participants, we summed *AIC* values across participants, $b$,

$$AIC_{sum} = \sum_{b=1}^{b} AIC_b \tag{12}$$

For a single participant, *BIC* is defined as:

$$BIC = -2\log(L) + z\log(n) \tag{13}$$

where $z$ is the number of model parameters, $\log(L)$ is the log likelihood of the dataset and $n$ is the number of trials in the dataset. Because *BIC* is sensitive to the total sample size used to compute the likelihood, calculating the summed *BIC* for each model requires that the sample size accounts for the total number of trials across all participants. We therefore summed the likelihood terms across participants, $b$, and added to that the total number of free parameters across participants, $zB$, multiplied by the log transformed total number of trials across participants, $nB$:

$$BIC_{sum} = \sum_{b=1}^{b} -2\log(L_b) + zB\log(nB) \tag{14}$$

where $B$ is the total number of participants.

## Visualisation of model fits

**Non-Bayesian models.** To visualise the fit of the non-Bayesian models, we generated a synthetic dataset of model responses using the best fitting parameters for each participant. Using the experimental data for that participant, for each trial we took 100 samples from a normal distribution with mean $s_i$ (the true stimulus value) and standard deviation $\sigma_I$ (estimated value of $\sigma$ for a given intensity $I$). We then compared each sample to the boundary parameters ($b_1 - b_3$ for the different means task and $b_1 - b_7$ for the different SDs task) to generate 100 category/confidence responses for that trial. We took the mean of the generated category/confidence responses to produce a single average predicted response for that trial.

**Bayesian models.** As with non-Bayesian models, for each trial, we took 100 samples from a normal distribution with mean $s_i$ and standard deviation $\sigma_I$. We transformed boundary parameters from probability space into perceptual space (using the relevant equation from the model specification section) and compared each stimulus sample to the resulting boundaries to generate a predicted response. This approach generated 100 category/confidence responses for each trial, and we took the mean of the generated category/confidence responses.

**Bayesian models with decision noise.** For Bayesian models with decision noise, we compared each stimulus sample to 100 random draws of the category/confidence boundaries. We sampled from a normal distribution with standard deviation $\sigma_d$ and mean $k$. Each draw was converted into perceptual space using the relevant equation and $\sigma_I$. We generated $100 \times 100$ category/confidence responses for each trial and took the mean of the generated category/confidence responses to produce a single average response for that trial.

**Generating figures.** For plots with stimulus value (or a transformation of stimulus value) on the horizontal axis, stimulus values were binned so that each bin consisted of the same number of trials. For visualisation, we calculated the mean category/confidence response across all participants for both the empirical data and model predictions for each bin.

### Best fitting model parameters

We report the best fitting parameters for each participant for each model variant in **S3 Table** (different means task), **S4 Table** (different SDs task) and **S5 Table** (control study; see **S1 Text** for more information).

## Supporting information

**S1 Text. Control Study. Fig A**. Control Experiment: Categorisation Accuracy as a Function of Stimulus Intensity. **Table A**. Model Fits for Control Data. **Table B**. Parameter Settings Across Modalities for Control Data. **Fig B**. Model Comparison for Control Study.
(DOCX)

**S2 Text. Secondary Model-Free GLMMs. Table A**. Category and Confidence Model-Free GLMMs With Data From All Tasks and Modalities. **Fig A**. Category and Confidence Model-Free GLMMs With Data From All Tasks and Modalities
(DOCX)

**S3 Text. Alternative Visualisation of Model Fits from Main Experiment. Fig A**. Unscaled-Evidence Strength Model: The Distance Model. **Fig B**. Scaled-Evidence Strength Model: The Linear Model. **Fig C**. Scaled-Evidence Strength Model: The Quadratic Model. **Fig D**. Scaled-Evidence Strength Model: The Free-Exponent Model with Orientation Dependent Noise. **Fig E**. Bayesian Model: Log Posterior Probability Ratio. **Fig F**. Bayesian Model: Log Posterior Probability Ratio with Orientation Dependent Noise. **Fig G**. Bayesian Model: Log Posterior Probability Ratio with Decision Noise. **Fig H**. Bayesian Model: Log Posterior Probability Ratio with Free Category Distributions Parameters. **Fig I**. Parameter Settings Across Modalities: Different Means Task. **Fig J**. Parameter Settings Across Modalities: Different SDs Task
(DOCX)

**S4 Text. Model Recovery. Fig A**. Model Recovery for Core Models. **Fig B**. Model Recovery for Models Used To Compare Parameter Settings Across Modalities
(DOCX)

**S5 Text. Parameter Recovery for Different SDs Task Models. Fig A**. Parameter Recovery for Distance Model. **Table A**. Correlations Between Generating and Recovered Parameters for Unscaled Evidence Strength Models. **Fig B**. Parameter Recovery for Standard Bayesian Model. **Fig C**. Parameter Recovery for Bayesian Model with Free Category Distribution Parameters. **Fig D**. Parameter Recovery for Bayesian Model with Orientation Dependent Noise. **Fig E**. Parameter Recovery for Bayesian Model with Decision Noise. **Table B**: Correlations Between Generating and Recovered Parameters for Bayesian Models. **Fig F**. Parameter Recovery for Linear Model. **Fig G**. Parameter Recovery for Quadratic Model. **Fig H**. Parameter Recovery for Free-Exponent Model. **Fig I**. Parameter Recovery for Free-Exponent Model with Orientation Dependent Noise. **Table C**. Correlations Between Generating and Recovered Parameters for Scaled Evidence Strength Models. **Fig J.** Parameter Recovery with Evenly Spaced Stimulus Values. **Table D**. Correlations Between Generating and Recovered Parameters for Models with Evenly Spaced Stimulus Values
(DOCX)

**S6 Text. Parameter Recovery for Cross-Modal Models. Fig A**. Parameter Recovery for Common Settings Model. **Fig B**. Parameter Recovery for Different Noise Settings Model. **Fig C.** Parameter Recovery for Flexible Settings Model. **Table A**. Correlations Between Generating and Recovered Parameters for Cross-Modal Models
(DOCX)

**S7 Text. Headphone Calibration.**
(DOCX)

**S8 Text. Derivation of d for Different Means and Different SDs Tasks.**
(DOCX)

**S9 Text. Model Fitting.**
(DOCX)

**S1 Table. Categorisation Accuracy for Highest Intensity Stimuli and Exclusions.**
(DOCX)

**S2 Table. GLMM Follow-Up Tests: The Effect of Evidence at Intensity Levels.**
(DOCX)

**S3 Table. Parameter Estimates for Different Means Task.**
(XLSX)

**S4 Table. Parameter Estimates for Different SDs Task.**
(XLSX)

**S5 Table. Parameter Estimates for Control Data.**
(XLSX)

## Acknowledgments

We would like to thank Dr David Lloyd and Associate Professor Wayne Wilson for their assistance with headphone calibration. We would also like to thank Dr William Adler for helpful comments and feedback about the model fitting.

## Author Contributions

**Conceptualization:** Rebecca K. West, Natasha Matthews, Jason B. Mattingley, David K. Sewell.

**Data curation:** Rebecca K. West.

**Formal analysis:** Rebecca K. West, William J. Harrison, David K. Sewell.

**Investigation:** Rebecca K. West.

**Methodology:** Rebecca K. West, William J. Harrison, Natasha Matthews, Jason B. Mattingley, David K. Sewell.

**Project administration:** Rebecca K. West, David K. Sewell.

**Resources:** Natasha Matthews, Jason B. Mattingley, David K. Sewell.

**Software:** Rebecca K. West.

**Supervision:** Natasha Matthews, Jason B. Mattingley, David K. Sewell.

**Validation:** Rebecca K. West.

**Visualization:** Rebecca K. West.

**Writing – original draft:** Rebecca K. West.

**Writing – review & editing:** Rebecca K. West, William J. Harrison, Natasha Matthews, Jason B. Mattingley, David K. Sewell.

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
