## [Decision Letter · Decision Letter 0]

22 Feb 2023

Dear Ms West,

Thank you very much for submitting your manuscript "Modality Independent or Modality Specific? Common Computations Underlie Confidence Judgements in Visual and Auditory Decisions" for consideration at PLOS Computational Biology.

As with all papers reviewed by the journal, your manuscript was reviewed by members of the editorial board and by several independent reviewers. In light of the reviews (below this email), we would like to invite the resubmission of a significantly-revised version that takes into account the reviewers' comments.

We cannot make any decision about publication until we have seen the revised manuscript and your response to the reviewers' comments. Your revised manuscript is also likely to be sent to reviewers for further evaluation.

Sincerely,

Christoph Mathys

Academic Editor

PLOS Computational Biology

Thomas Serre

Section Editor

PLOS Computational Biology

Reviewer's Responses to Questions

**Comments to the Authors:**

Reviewer #1: The authors investigate through modeling fitting several computational models for confidence judgements, and whether they generalize to both visual and auditory tasks. The authors find that scaled evidence strength models outperform unscaled and Bayesian models. The paper builds over previous literature on similar experiments but adds novelty by introducing the study of two modalities and an exhaustive model comparison which is of value.

My main criticism on this paper is at the interpretative side: the authors seem to imply, based on the model fitting, that humans are non-Bayesian. But this conclusion totally depends on the modelling assumptions. I think that the authors make a good job in comparing different models quite exhaustively but concluding that “it is unlikely that confidence judgements…. are generated by a Bayesian computational process” can be too much of a stretch. This is so because things like priors in the Bayesian model, or different noise models (e.g., non Gaussian) could account for the departures of the model with the data (see the relevant paper of Schustek et al, Nature Comm, 2019). Also, the scaled model is not mechanistic nor normative, so while Bayesian models could be taken to generalize to other domains, the mechanistic model might be able to do so. I think that the authors should restate the implications of their results in view of this criticism and answer the question of what could be the reason for having scaling evidence strength with linear of quadratic dependences.

Second, the models assume that there is a point estimate of uncertainty. But it is probably that in the Bayesian model agents need to consider also a distribution over uncertainty. Such a distribution over uncertainty could be an additional free parameter that could effectively change the Gaussian noise distribution to another one with a longer tail. Does such a model make a better job in fitting the data?

In the list of refs in line 155, the paper by Moreno, Neural Computation, 2010 on confidence judgments is missing. Also, the paper by Drugowitsch lab, Drugowitsch et al, J of Neuroscience, 2012, seems to be relevant and earlier to those quoted. Regarding the first paper, do the authors recorded the reaction times? If yes, one could model drift-diffusion models to the data. The Bayesian models does not only predict how accuracy and confidence depends on the stimulus strength (drift rate) and uncertainty (inverse noise variance), but also on the reaction time. A discussion of these effects could be relevant if the data are available, or in the Discussion if they are not available in the current experiment.

Methodologically, it is convenient to use 4 confidence ratings for the model fitting, but it does not seem to be the most natural way of conveying confidence, which can be made numerically by the participants using the notion of probability of being correct. It could be relevant to describe in the Methods the pros and cons of this choice.

While there is overlap of the category distributions in the visual task, in the auditory task the overlap seems to be minor (comparing the mean distance and the standard deviations). It is unclear the reason for this choice, so a clarification could be beneficial.

In line 391, it is unclear whether b1 lies in between b2 and -b2, or can be independently fitted.

In Fig. 4, it is unclear what point corresponds to. Is it an average across participants? It would be important to clarify this, and maybe also add error bars.

In the author summary, I would more clearly define “a single class of models”, and “governed by the same general process”. This wording is largely ambiguous and unspecific.

The terminology “response” in Fig. 4 can be confusing, as it adds to confidence and category responses, used as well.

Minor comments:

Some papers listed in the paper do not show in the reference list, as far as I can tell.

There seems to be some problem in the pdf version with how the gratings are displayed in Fig. 1A and 1B.

In line 501, redefining d could add clarity in this passage.

Reviewer #2: This is comprehensive and careful work examining the extent to which different computational models of confidence generalise across visual and auditory modalities. Such a computational approach has great promise to shed light on the extent to which metacognition is domain-general or domain-specific, which has been much debated in the literature but has often only focused on summary statistics such as meta-d’ rather than a detailed consideration of the model that generates confidence data. Here, this is tackled with 2 complementary tasks that were matched across modalities, and in-depth psychophysical characterisation of confidence thresholds across 8 observers.

I am enthusiastic about this paper which I think should be published following revisions. I list a number of specific comments below.

1) The analysis and model comparison is comprehensive, and I appreciate the careful work that has gone into the parameter recovery simulations reported in the Supplement. However, at a number of points in the manuscript I found myself wondering how identifiable a specific model is. For instance, the scaled evidence-quadratic and Bayesian models in Figure 3 look very similar in terms of their predictions for confidence thresholds. Model recovery can be easily established using a ground-truth simulation, in which data from each model is simulated and then fit with each candidate model, to generate a confusion matrix (see eg https://elifesciences.org/articles/49547). In addition, it seems important to establish to what extent a model in which parameters are shared / flexible across two candidate modalities can be accurately recovered, as a worry in the last section is that the domain-specific model is preferred due to this additional flexibility.

2) In Table 1, the model-free GLM, were there significant interactions of predictors with task? Such an analysis would provide a model-free way to ask if performance and/or confidence profiles differed between the two modalities, prior to fitting of candidate models.

3) Is there a reason that the different-means task for audition exhibits apparently binary dependence on intensity (eg in Figure 5)? It looks like intensity values 1/2 and 3/4 cluster together, whereas in vision the influence of intensity is more graded.

4) On the potential limitations of Bayesian models – I wondered whether allowing for uncertainty over stimulus uncertainty would provide additional help here, especially given that part of the failure in the Bayesian model fits appears to be a miscalibration of confidence with respect to (performance-dependent) fitted uncertainty. I am not suggesting adding another model to the mix, but the authors might wish to consider this in Discussion, in relation to this paper: https://www.nature.com/articles/s41562-022-01464-x

Reviewer #3: The manuscript is a report of four psychophysical experiments and extensive computational modelling of the ability of human participants to judge their confidence in the correctness of their perceptual decisions. The psychophysical experiments cover two binary categorical tasks, one where stimuli were drawn from two distributions that had different means but same variance, and the other where stimuli were drawn from two distributions that had same mean but different variances. These two tasks were run in both the visual and the auditory sensory modalities. The computational modelling covers three different classes of models, the first where the strength of their sensory evidence is taken directly to inform confidence evidence, the second where the sensory evidence is scaled by some factor that can depend on the intensity of the sensory signal, and the third based on a Bayesian account of the tasks. The authors find that the scaled evidence models were better models overall, for both sensory modalities, and especially for the different variances tasks. The manuscript is extremely well written, with an acute sense for clarity in the description of the results and models. The analyses are trustworthy and the comparison across tasks, sensory modalities, and model classes is particularly welcome in the growing literature on perceptual confidence. I have just a few comments.

First, I fail to understand how the unscaled evidence strength model can work across sensory modalities (see description on lines 438ff). Since the sensory evidence is a noisy bias-free estimate of the sensory stimulus (Equation 1, page 18), this evidence has the unit of the independent variable (i.e. degrees for the visual task, Hertz for the auditory task). So, how can one place the same perceptual boundaries ‘b_r’ for both the visual and the auditory tasks? The same question arises for the scaled evidence model with common settings (see lines 467ff). Was this possible because the participants data were normalized in some way? This may be what the authors call “standardised stimulus values” (page 38, line 763). I think the authors should try to clarify this point.

Second, it is not clear to me how the evidence for one category and the category diagnosticity variables were computed (see Figures 4B and 4C). These variables are defined in Equations 12 and 13, but these equations are based on the knowledge of ‘x’, but my understanding is that ‘x’ is a noisy measurement at each trial (see page 18, line 342), and thus it is an unknown internal variable. Did the authors use the average noisy measurement consistent with the participant’s categorical response and confidence judgment?

Third, I think it would be interesting to report the values of some of the fitted parameters, at least for the winning model (averaged across participants or for one typical participant). In particular, for the scaled evidence model, it would be informative to compare the scaling of the visual and auditory sensory evidence, in order to get an appreciation of the efficiency of the visual and auditory confidence computations.

Minor comments:

- page 1: it seems that there is a typo in the email address of the corresponding author (missing one ‘c’).

- the manuscript sometimes refers to Locke et al. (2021) when I think they mean 2022 (e.g. lines 139, 175).

- Figure 2B: personally I would have matched the y-range in the bottom panel to that of the top panel (i.e. between 0 and 1) to emphasize that the evidence for category 2 in the different SDs task does not go down to zero.

- page 17: I understand that the different categories and intensity levels were all interleaved within a block of trials, rather than having a single intensity level (and both categories) in some mini-blocks. Maybe this could be clarified in the methods section.

- page 25, line 509: the parameter ‘mu_2’ does not appear in the equations above. Is this because in Equation 3, ‘mu_2’ = - ‘mu_1’? Also, it would probably be nice here to clearly state the difference between ‘sigma’ and ‘sigma_1’.

- page 28, line 571: the parameter t that refers to the total number of trials here was used before (page 24, line 489) to refer to the sensory modality.

- page 29, Equations 9 and 11: there is a missing ‘B’ on top of the sum sign.

- Figure 5: can the authors match the range of y-axis across all panels, like what they have done for Figure 4?

- Figures 7A and 7B: why not remove the circular symbols, that are redundant with the solid lines, so that the square symbols can be more visible?

- Figures 7C and 7D: from the text (page 50, lines 996 and 997), it looks like the labels “offset noise” and “different noise” have been inverted.

- Supplementary information, line 247: I think “confidence” is on the y-axis, not the x-axis.

**Have the authors made all data and (if applicable) computational code underlying the findings in their manuscript fully available?**

Reviewer #1: Yes

Reviewer #2: None

Reviewer #3: Yes

PLOS authors have the option to publish the peer review history of their article (what does this mean?). If published, this will include your full peer review and any attached files.

Reviewer #1: No

Reviewer #2: No

Reviewer #3: **Yes: **Pascal Mamassian
---

## [Decision Letter · Decision Letter 1]

6 Jun 2023

Dear Ms West,

We are pleased to inform you that your manuscript 'Modality Independent or Modality Specific? Common Computations Underlie Confidence Judgements in Visual and Auditory Decisions' has been provisionally accepted for publication in PLOS Computational Biology.

Best regards,

Christoph Mathys

Academic Editor

PLOS Computational Biology

Thomas Serre

Section Editor

PLOS Computational Biology

Reviewer's Responses to Questions

**Comments to the Authors:**

Reviewer #1: The authors have appropriately addressed all my comments and concerns.

Reviewer #2: The authors have conducted a comprehensive and careful revision. I have no further comments.

Reviewer #3: I thank the authors for taking all the comments of all reviewers seriously. I do not have any outstanding comment.

**Have the authors made all data and (if applicable) computational code underlying the findings in their manuscript fully available?**

Reviewer #1: Yes

Reviewer #2: None

Reviewer #3: None

PLOS authors have the option to publish the peer review history of their article (what does this mean?). If published, this will include your full peer review and any attached files.

Reviewer #1: No

Reviewer #2: No

Reviewer #3: **Yes: **Pascal Mamassian

---

## [Editor Report · Acceptance letter]

11 Jul 2023

PCOMPBIOL-D-22-01598R1 

Modality independent or modality specific? Common computations underlie confidence judgements in visual and auditory decisions

Dear Dr West,

I am pleased to inform you that your manuscript has been formally accepted for publication in PLOS Computational Biology. Your manuscript is now with our production department and you will be notified of the publication date in due course.

With kind regards,

Zsofi Zombor
